# Comparative safety and efficacy of cognitive enhancers for Alzheimer's dementia: a systematic review with individual patient data network meta-analysis

Areti Angeliki Veroniki ,[1,2] Huda M Ashoor,[2] Patricia Rios,[2] Georgios Seitidis ,[3] Lesley Stewart,[4] Mike Clarke ,[5] Catrin Tudur-Smith,[6] Dimitris Mavridis ,[3] Brenda R Hemmelgarn,[7] Jayna Holroyd-Leduc,[8] Sharon E Straus,[2,9] Andrea C Tricco [2,10]

For numbered affiliations see end of article.

**Correspondence to**
Dr Areti Angeliki Veroniki;
Areti-Angeliki.Veroniki@unityhealth.to

## ABSTRACT

**Objective** To examine the comparative efficacy and safety of cognitive enhancers by patient characteristics for managing Alzheimer's dementia (AD).

**Design** Systematic review and individual patient data (IPD) network meta-analysis (NMA) based on our previously published systematic review and aggregate data NMA.

**Data sources** MEDLINE, Embase, Cochrane Methodology Register, CINAHL, AgeLine and Cochrane Central Register of Controlled Trials up to March 2016.

**Participants** 80 randomised controlled trials (RCTs) including 21 138 adults with AD, and 12 RCTs with IPD including 6906 patients.

**Interventions** Cognitive enhancers (donepezil, rivastigmine, galantamine and memantine) alone or in any combination against other cognitive enhancers or placebo.

**Data extraction and synthesis** We requested IPD from authors, sponsors and data sharing platforms. When IPD were not available, we used aggregate data. We appraised study quality with the Cochrane risk-of-bias. We conducted a two-stage random-effects IPD-NMA, and assessed their findings using CINeMA (Confidence in Network Meta-Analysis).

**Primary and secondary outcomes** We included trials assessing cognition with the Mini-Mental State Examination (MMSE), and adverse events.

**Results** Our IPD-NMA compared nine treatments (including placebo). Donepezil (mean difference (MD)=1.41, 95% CI: 0.51 to 2.32) and donepezil +memantine (MD=2.57, 95% CI: 0.07 to 5.07) improved MMSE score (56 RCTs, 11 619 participants; CINeMA score: moderate) compared with placebo. According to P-score, oral rivastigmine (OR=1.26, 95% CI: 0.82 to 1.94, P-score=16%) and donepezil (OR=1.08, 95% CI: 0.87 to 1.35, P-score=30%) had the least favourable safety profile, but none of the estimated treatment effects were sufficiently precise when compared with placebo (45 RCTs, 15 649 patients; CINeMA score: moderate to high). For moderate-to-severe impairment, donepezil, memantine and their combination performed

## Strengths and limitations of this study

► This is one of the most comprehensive systematic reviews and network meta-analysis of cognitive enhancers including individual patient data for Alzheimer's dementia to produce treatment recommendations by patient characteristics.

► We followed the methodologically rigorous guidelines in the Cochrane Handbook for systematic reviews, and assessed credibility in the results using the Confidence in Network Meta-Analysis tool.

► Access to individual patient data allowed us to (1) observe minor differences between the original published results and our reanalysis, potentially due to differences in imputation methods for missing data or because original studies have excluded some patients, and hence have used a smaller sample size, (2) overcome potential reporting bias and (3) assess for potential effect modifiers that were not reported in the original publications (eg, comorbidities, additional medications) and explore for treatment-by-covariate interactions on the patient-level.

► Two-thirds of the included randomised controlled trials (RCTs), were associated with high risk of bias for incomplete outcome data due to attrition.

► We were unable to include individual patient data for all RCTs (only 15% of the studies shared their individual patient data), highlighting potential retrieval bias.

► Our literature searches were conducted 5 years ago and additional relevant studies may be available. However, obtaining individual patient data in a timely manner was very challenging and required more time than anticipated. Similar to all systematic reviews, the evidence should be updated regularly.

best, but for mild-to-moderate impairment donepezil and transdermal rivastigmine ranked best. Adjusting for MMSE baseline differences, oral rivastigmine and galantamine

improved MMSE score, whereas when adjusting for comorbidities only oral rivastigmine was effective.

**Conclusions** The choice among the different cognitive enhancers may depend on patient's characteristics. The MDs of all cognitive enhancer regimens except for single-agent oral rivastigmine, galantamine and memantine, against placebo were clinically important for cognition (MD larger than 1.40 MMSE points), but results were quite imprecise. However, two-thirds of the published RCTs were associated with high risk of bias for incomplete outcome data, and IPD were only available for 15% of the included RCTs.

**PROSPERO registration number** CRD42015023507.

## INTRODUCTION

Alzheimer's dementia (AD) is the most common type of dementia.[1] Patients living with AD have a lower quality of life due to deterioration in function, cognition, behaviour and mental health over time, as well as increased mortality.[2] Pharmacological treatment for AD predominantly consists of cholinesterase inhibitors (donepezil, galantamine, rivastigmine) and the N-methyl-d-aspartate receptor antagonist, memantine. All three cholinesterase inhibitors and memantine are currently the only effective licenced treatments for dementia,[3] but their clinical effect can be small and there is no convincing evidence that they modify the disease process in AD.[4] Also, it is unclear whether galantamine, rivastigmine or donepezil should be used by patients with severe AD, or whether memantine is the optimal treatment for severe AD.[5]

In AD, disease severity and sex are potential effect modifiers. However, aggregate data and covariates of interest (eg, sex, disease severity) are not consistently reported across randomised clinical trials (RCTs).[6] The use of individual patient data (IPD) has several advantages, such as it allows for the exploration of the relationship between treatment effects and patient-level characteristics, and it overcomes restrictions in using the information reported in the publication among others. The aim of this study was to examine the comparative efficacy and safety of cognitive enhancers for patients with different characteristics, such as severities of AD and for women versus men through a systematic review and IPD network meta-analysis (NMA). This systematic review was based on our previously published systematic review and aggregate data NMA.[6] NMA is an extension of standard meta-analysis synthesising different sources of evidence from a network of RCTs comparing different treatments within a single model. NMA can provide treatment effect estimates for treatment comparisons that have not studied in a head-to-head study.

## METHODS

We reported our results according to the Preferred Items for Systematic Reviews and Meta-Analysis (PRISMA) statement for NMA and PRISMA-IPD.[7 8]

### Protocol

The research question and protocol were based on our previous systematic review and NMA.[6] We registered our systematic review protocol with the prospective register of systematic reviews (PROSPERO), and published our protocol.[9] Additional information is also provided in online supplemental appendix 1 and online supplemental file 2. Herein, we briefly summarise our methods.

### Eligibility criteria

We updated our previous systematic review,[6] using similar population, interventions, comparators, study designs and time period criteria. The literature search was updated from January 2015 to March 2016. We included published and English RCTs that assessed cognition via the Mini-Mental State Examination (MMSE; efficacy and primary outcome) and/or adverse events (AE; safety outcome) in adults with AD.

### IPD collection process

We contacted the corresponding author followed by the next-in-order author, as presented in each eligible RCT, to obtain IPD. The author contact process was part of an RCT that our team conducted to assess methods that may optimise response rates for IPD retrieval.[10] We also contacted sponsors of eligible trials, as reported in the publications. We contacted industry sponsors only, as we were not able to locate contact information for non-industry sponsors (eg, grants and university funding). If a study had multiple sponsors, we contacted all of them. To further facilitate IPD access, we contacted the Clinical Study Data Request[11] and Yale University Open Data Access data sharing platforms.[12] If a data provider was unable to provide IPD we noted the reason.

### Risk of bias and quality appraisal

We appraised study quality using the Cochrane risk of bias tool.[13] To ensure data consistency[8] we compared IPD with aggregate data reported in the publication. We assessed whether randomisation of patients was adequate (ie, intervention and comparison groups were balanced for important patient characteristics), by comparing numbers and types of patients in each arm.

When at least 10 studies were available for each treatment against placebo, publication bias and small-study effects were examined visually using the comparison adjusted funnel plot under the fixed-effect model.[3] When a funnel plot asymmetry was detected, we performed the Copas selection for the treatment comparisons that were informed by at least 10 studies and for which asymmetry was evident in the funnel plot. We explored the possibility that this was due to publication bias,[14] and made moderate assumptions about the probability of publication of the smaller and larger (in terms of SE) studies. We assumed that the smallest study had a probability of publication equal to 40%–50% and the largest study had a probability of 80%–90%. Confidence in NMA findings was assessed for each outcome using CINeMA (Confidence in Network

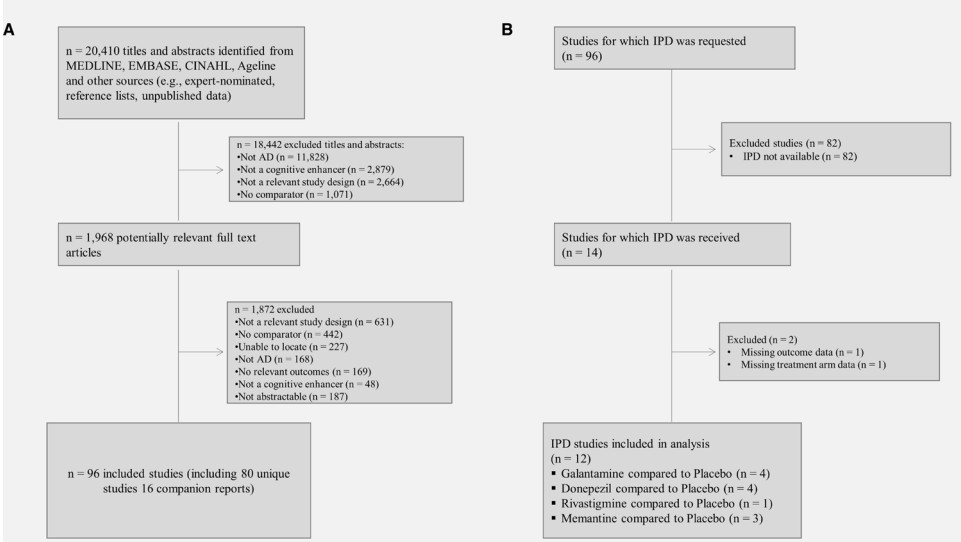

**Figure 1** Flow diagram for study inclusion in the review (A) and studies retrieved with individual patient data (B). AD, Alzheimer's dementia; IPD, individual patient data.

Meta-Analysis, see online supplemental appendix 1 for more details).[15]

## Synthesis

We performed a descriptive analysis using frequencies and distributions of the characteristics of the included patients and treatments. For each outcome, we present the network geometry according to IPD availability. We conducted a two-stage IPD analysis, whereby data were analysed separately in each trial in the first stage and the trial parameter estimates were synthesised in a random-effects meta-analysis or NMA in the second stage.

The summary treatment effects are presented using the OR or mean difference (MD) along with their corresponding CIs and prediction intervals (PIs).[16] We ranked the interventions for each outcome using the P-scores (and SUCRAs (surface under the cumulative ranking curve) in meta-regression analysis), and present them in a rank-heat plot.[17 18]

## Patient and public involvement

Not applicable.

## RESULTS

### Literature search, study selection and IPD obtained

After screening 20 410 titles and abstracts and 1968 full-text articles, 96 studies fulfilled the eligibility criteria; 80 unique studies and 16 companion reports (figure 1A, online supplemental appendix 2).

Of the 80 unique RCTs, 55 reported at least one industry-sponsored funder (ie, 40 studies reported a single industry-sponsor and 15 multiple industry-sponsors). In the remaining studies, nine were publicly-sponsored and 16 did not report any information about funding. We requested IPD by contacting the corresponding authors for 80 RCTs that included 21 138 participants. None of

the original authors shared their IPD. Fifteen commercial sponsors were then contacted and 6 (40%) sponsors shared their data through proprietary sponsor-specific platforms. The six sponsors were contacted for 46 RCTs (14 580 participants), and we obtained IPD for 30% (14 RCTs, 8007 participants) of these RCTs (1058 total waiting days up to 9 March 2020). The study flow for obtaining IPD is depicted in figure 1B.

We were able to include 12 (6906 patients) of 14 RCTs in our NMA due to incompleteness of provided IPD (online supplemental appendix 3). The number of studies with available/non-available IPD from each data provider along with reasons for non-availability of IPD are presented in online supplemental appendix 4.

### Study and patient characteristics

Most included studies (33%) were multinational. The mean age of patients ranged from 61 to 86 years. The majority of the RCTs included patients with mild–moderate AD (55%), although the diagnostic criteria used for AD varied widely table 1. The most frequent longest duration of follow-up was 24 weeks (24 RCTs, 30%; online supplemental appendix 5). Important patient characteristics, such as per cent of men and dropout rates, were not balanced across groups in the RCTs with provided IPD (online supplemental appendix 6). Comparing study and patient characteristics of available and non-available IPD when a study was industry-sponsored, we found differences in the year of study publication, study size and absolute MD (online supplemental appendix 7).

### Risk of bias and IPD integrity

Using the Cochrane risk-of-bias tool, allocation concealment was at low risk of bias for 43% and blinding of participants and personnel was low for 64% of the RCTs (online supplemental appendix 8). One-third of the RCTs had

**Table 1** Study and patient characteristics

| | AD (N=80) | IPD (N=12) |
|---|---|---|
| Total number of participants | 21 138 | 6906 |
| Longest duration of follow-up in weeks: mean (range) | 28.28 (8.00–208.00) | 29.33 (12.00–104.00) |
| Mean number of patients (range) | 264 (14–2045) | 4867 (123–2045) |
| Mean age in years (range) | 74.64 (61.00–85.70) | 73.94 (70.40–78.00) |
| Mean % female (range) | 61.35 (3.00–89.00) | 62.76 (53.68–81.00) |
| Country of conduct: frequency (%) | | |
| Canada | 2 (2.50) | 1 (8.33) |
| China | 6 (7.50) | – |
| Germany | 1 (1.25) | – |
| Iran | 2 (2.50) | – |
| Italy | 6 (7.50) | – |
| Japan | 7 (8.75) | 1 (8.33) |
| Norway | 1 (1.25) | – |
| Romania | 1 (1.25) | – |
| South Korea | 1 (1.25) | – |
| Spain | 3 (3.75) | – |
| Sweden | 2 (2.50) | – |
| Turkey | 1 (1.25) | – |
| UK | 6 (7.50) | 1 (8.33) |
| USA | 15 (18.75) | – |
| Multinational | 26 (32.50) | 9 (75.00) |
| Interventions examined: frequency* | | |
| Placebo/no treatment | 61 (76.25) | 12 (100.00) |
| Donepezil | 47 (58.75) | 4 (33.33) |
| Galantamine | 20 (25.00) | 4 (33.33) |
| Memantine | 20 (25.00) | 3 (25.00) |
| Rivastigmine† | 18 (22.50) | 1 (8.33) |
| Outcomes reported: frequency* | | |
| Mini-Mental State Examination | 57 (71.25) | 6 (50.00) |
| Adverse events | 46 (57.50) | 12 (100.00) |
| Funding | | |
| Industry-sponsored | 48 (60.00) | 12 (100.00) |
| Publicly-sponsored‡ | 9 (11.25) | – |
| Mixed | 7 (8.75) | – |
| Not reported | 16 (20.0) | – |
| Severity of AD: frequency (%) | | |
| Mild | 3 (3.75) | – |
| Mild–moderate | 44 (55.00) | 7 (58.33) |
| Mild–severe | 2 (2.50) | – |
| Moderate | 3 (3.75) | – |
| Moderate–severe | 11 (13.75) | 1 (8.33) |
| Severe | 6 (7.50) | 2 (16.67) |
| Not reported | 11 (13.75) | 2 (16.67) |
| Diagnostic criteria for AD: frequency* | | |

Continued

**Table 1** Continued

| | AD (N=80) | IPD (N=12) |
|---|---|---|
| Mini-Mental State Examination | 70 (87.50) | 12 (100.00) |
| National Institute of Neurological Disorders and Stroke-Alzheimer Disease and Related Disorders Association | 67 (83.75) | 12 (100.00) |
| Diagnostic and Statistical Manual of Mental Disorders | 39 (48.75) | 5 (41.67) |
| MRI/CT | 9 (11.25) | 2 (16.67) |
| Clinical Dementia Rating | 6 (7.50) | – |
| Hachinski Ischemic Score | 5 (6.25) | – |
| Alzheimer's Disease Assessment Scale-Cognitive Subscale | 3 (3.75) | 1 (8.33) |
| Other | 20 (25.00) | 1 (8.33) |

*Multiple interventions and outcomes reported per study.
†Rivastigmine refers to either oral or transdermal administration.
‡Including sponsors such as the National Institute of Aging, UK Medical Research Council and Veteran Affairs.
–, not applicable; AD, Alzheimer's dementia ; IPD, individual patient data .

low risk of incomplete outcome data bias due to attrition and almost two-thirds had high potential risk of 'other' bias, specifically, funding bias. The other risk of bias item was scored as unclear for 32%. Overall risk of bias was comparable in studies with available and unavailable IPD (online supplemental appendix 9).

All IPD provided were checked for consistency and results from published RCTs were reproduced and provided in online supplemental appendix 10. High dropout rates were observed in the IPD; experiencing an AE was the most common reason for dropout. Despite the high dropout rates observed in the individual studies, there was no indication of correlation between age and dropout (online supplemental appendix 11). Comparison-adjusted funnel plot for MMSE suggested there is indication for small-study effects (see online supplemental appendix 12). In contrast to the standard meta-analysis (MD=1.65, 95% CI: (0.16 to 3.14)), the Copas selection model estimated a pooled treatment effect for donepezil versus placebo (MD=1.87, 95% CI: (1.55 to 2.20)) with between-study variance $\tau^2$=1.95, and correlation coefficient −0.45 (−0.76 to −0.01) reflecting the belief that the propensity for publication was associated with the observed effect size.

## NMA

In both MMSE and AE outcomes, on average there were no important concerns regarding the transitivity and consistency assumptions (online supplemental appendices 13 and 14; design-by-treatment interaction test MMSE: $\chi^2$=4.36, 13 df, p value=0.987; AE: $\chi^2$=3.57, 6 df, p value=0.735). Below we present the main analysis results compared with placebo. Additional analyses are presented in online supplemental appendices 15 and 16. The network geometry is presented in figure 2.

### Cognition

The NMA for MMSE included 56 RCTs, 9 treatments (including placebo) and 11 619 participants. Nine RCTs (3625 patients) contributed IPD and 47 RCTs (7994 patients) contributed aggregated data to the NMA. Two studies[19 20] did not report MMSE in the final publication, but in the retrieved IPD we were able to use data for this outcome.

### NMA of studies with IPD and aggregate data

Studies in this NMA compared all available treatments. Donepezil (MD=1.41, 95% CI: 0.51 to 2.32) and donepezil +memantine (MD=2.57, 95% CI: 0.07 to 5.07) were superior to placebo in terms of MMSE score (online supplemental appendix 15). Transdermal rivastigmine

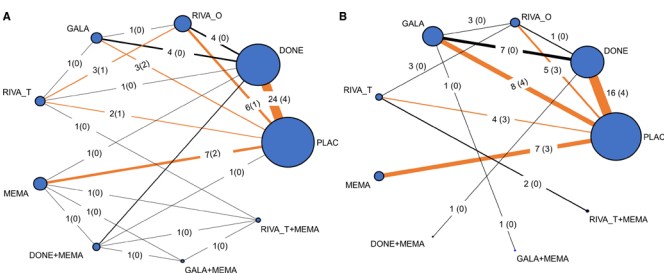

**Figure 2** Network diagrams for (A) MMSE and (B) AE outcomes. The size of each node and line indicates the number of studies included in each treatment comparison. The number of studies per treatment comparison is presented on each edge, and the number of studies with individual patient data (IPD) is depicted in a parenthesis. Orange coloured edges are informed by both IPD and aggregate data, whereas black coloured edges are informed by aggregate data only. AE, adverse event; DONE, donepezil; GALA, galantamine; MEMA, memantine; MMSE, Mini-Mental State Examination; PLAC, placebo; RIVA_O, oral rivastigmine; RIVA_T, transdermal rivastigmine.

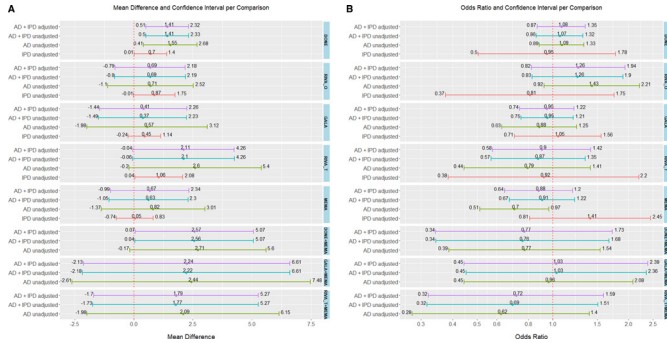

**Figure 3** Forest plot of network meta-analysis (NMA) results for all cognitive enhancers versus placebo in (A) MMSE outcome, and (B) AE outcome. NMA results are presented for (i) aggregate data (AD) and fully adjusted results from studies with available individual patient data (IPD), (ii) AD and crude results from studies with available IPD, (iii) AD only (studies with available IPD are not included in the analysis) and (iv) crude results from individual studies with IPD. AD, Alzheimer's dementia; AE, adverse events; DONE, donepezil; GALA, galantamine; MEMA, memantine; MMSE, Mini-Mental State Examination; PLAC, placebo; RIVA_O, oral rivastigmine; RIVA_T, transdermal rivastigmine.

(MD=2.11, 95% CI: −0.04 to 4.26), and the combinations donepezil +memantine, galantamine +memantine (MD=2.24, 95% CI: −2.13 to 6.61), and transdermal rivastigmine +memantine (MD=1.79, 95% CI: −1.70 to 5.27) were associated with a MD from placebo of more than 1.40 MMSE points. A previous study suggested a MD larger than 1.40 is a minimal clinically important difference (MCID).[21] However, the associated 95% CIs were quite imprecise spanning between a mean decrease below and a mean increase above the suggested MCID value (figure 3A). However, donepezil +memantine had the highest likelihood of being the most effective in improving MMSE score (P-score range 79%–80%, figure 4). Confidence in NMA results was moderate (online supplemental appendix 17).

### NMA of studies with aggregate data
Studies in this NMA compared all available treatments. Donepezil improved MMSE score significantly (MD=1.55, 95% CI: 0.41 to 2.68). Assuming an MCID of 1.40, results were in agreement with the NMA of IPD and aggregate data, and donepezil +memantine (MD=2.71, 95% CI: −0.17 to 5.60) was likely the most effective in improving MMSE score (P-score=76%).

### NMA of studies with IPD
Studies in this NMA compared placebo, donepezil, oral rivastigmine, transdermal rivastigmine, galantamine and memantine. Donepezil (MD=0.70, 95% CI: 0.01 to 1.40) and transdermal rivastigmine (MD=1.06, 95% CI: 0.04 to 2.08) were superior to placebo, but none of the point estimates reached a previously suggested MCID.[21] The most effective treatment was likely transdermal rivastigmine (P-score=82%).

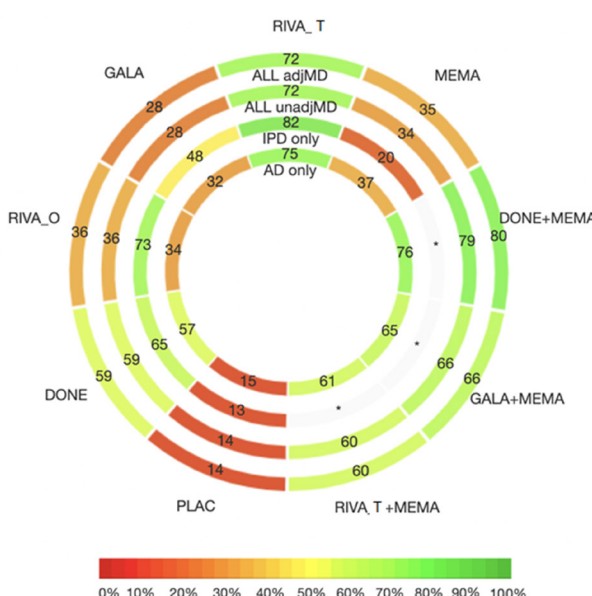

**Figure 4** Rank-heat plot of P-scores for nine treatments, including placebo, studied in randomised clinical trials with patients with Alzheimer's dementia assessing Mini-Mental State Examination. Circles from inside out present results for different network meta-analyses including: (i) aggregate data (AD) only (studies with available IPD are not included in the analysis), (ii) crude results from individual studies with individual patient data (IPD), (iii) AD and crude results from studies with available IPD and (iv) AD and fully adjusted results from studies with available IPD. Numbers within each sector correspond to the P-score values as calculated in each model. AD, Alzheimer's dementia; adjMD, adjusted mean difference; DONE, donepezil; GALA, galantamine; MEMA, memantine; PLAC, placebo; RIVA_O, oral rivastigmine; RIVA_T, transdermal rivastigmine; unadjMD, unadjusted MD.

### Additional analyses using IPD and aggregate data
Overall, additional analyses using both IPD and aggregate data were in agreement with the findings of the main analysis (online supplemental appendix 16). Cognitive performance was better in patients with mild-to-moderate MMSE receiving donepezil (MD=1.68, 95% CI: 0.31 to 3.06, P-score=69%) and most likely when receiving transdermal rivastigmine (MD=2.74, 95% CI: −0.68 to 6.16, P-score=81%). In patients with moderate-to-severe MMSE the combination donepezil +memantine improved MMSE score significantly (MD=2.49, 95% CI: 1.55 to 3.44, P-score=100%), but oral rivastigmine deteriorated MMSE score significantly (MD= −1.00, 95% CI: −1.87 to −0.12, P-score=4%). Donepezil (MD=1.31, 95% CI: 0.66 to 1.96, P-score=78%) and memantine (MD=0.69, 95% CI: 0.07 to 1.31, P-score=59%) also performed well for patients with moderate-to-severe cognitive impairment.

Accounting for the impact of the outlier studies, galantamine +memantine was the second-best cognitive enhancer (MD=1.87, 95% CI: 0.08 to 3.66, P-score=82%) after donepezil +memantine (MD=2.04, 95% CI: 1.03 to 3.05, P-score=92%). Using only IPD adjusted

for comorbidities suggested that oral rivastigmine improves MMSE score (MD=0.88, 95% CI: 0.31 to 1.45, P-score=75%). Similarly, using IPD adjusted for cognitive impairment assessed with MMSE at baseline suggested that oral rivastigmine (MD=0.88, 95% CI: 0.31 to 1.45, P-score=69%) and galantamine (MD=0.76, 95% CI: 0.34 to 1.18, P-score=62%) improve MMSE score, but in a future study, results are only stable for galantamine.

Heterogeneity in NMA was high (between-study variance=5.75, $I^2$=96%) compared also to the Rhodes *et al*[22] empirical distribution (median 0.05, 95% range: 0.00–7.56). However, heterogeneity decreased importantly when excluding outliers (between-study variance=0.59, $I^2$=73%), including only patients with moderate-to-severe AD (between-study variance=0.18, I2=44%), restricting to industry-sponsored trials (between-study variance=0.16, $I^2$=43%) and using IPD only (between-study variance=0.12, $I^2$=29%).

### Adverse events

An NMA was conducted on AEs (study definitions are provided in online supplemental appendix 18) with 45 RCTs, 9 treatments (including placebo) and 15 649 patients (figure 2B). In particular, 12 RCTs (6420 patients) contributed to the NMA using their IPD and 33 RCTs (9229 patients) using their data on their aggregated form. The time taken to achieve at least one AE was available in eight studies with available IPD and ranged between 45 and 2228 days (online supplemental appendix 19). Only one study included a patient with an AE occurring earlier than the trial opening and was excluded from the study.[23]

#### *NMA of studies with IPD and aggregate data*

Studies in this NMA compared all available treatments. According to P-score, oral rivastigmine had the least favourable safety profile regarding AE (OR=1.26, 95% CI: 0.82 to 1.94, P-score=16%), followed by donepezil (OR=1.08, 95% CI: 0.87 to 1.35, P-score=30%) and galantamine +memantine (OR=1.03, 95% CI: 0.45 to 2.39, P-score=43%), yet in these comparisons the odds of experiencing an AE were imprecise and not importantly different from placebo (figure 3b; online supplemental appendices 16 and 20). Confidence in NMA results ranged between moderate and high (online supplemental appendix 17).

#### *NMA of studies with aggregate data*

Studies in this NMA compared all available treatments. Results were mainly consistent with NMA of IPD and aggregate data, but memantine was 0.70 times less likely to experience an AE than placebo, with an OR ranging from 0.51 to 0.97 (P-score=77%).

#### *NMA of studies with IPD*

Studies in this NMA compared placebo, donepezil, oral rivastigmine, transdermal rivastigmine, galantamine and memantine. Results were on average consistent with NMA of IPD and aggregate data.

#### *Additional analyses using IPD and aggregate data*

Additional analyses using both IPD and aggregate data, showed that memantine was 0.61 times less likely to experience an AE than placebo when using study duration as a covariate, with an OR ranging from 0.37 to 0.93 (P-score=88%). Restricting to low risk of bias for incomplete outcome data, galantamine was associated with significantly lower odds of an AE (OR=0.69, 95% CI: 0.50 to 0.97, P-score=80%).

Heterogeneity in NMA was low (between-study variance=0.04, $I^2$=22%) compared with the Turner *et al*[24] empirical distribution (median 0.12, 95% range: 0.01– 2.63). Heterogeneity decreased importantly when restricting to aggregate data (between-study variance=0.00, $I^2$=0%), low risk of bias for incomplete outcome data (between-study variance=0.02, $I^2$=10%), patients with moderate-to-severe cognitive impairment (between-study variance=0.00, $I^2$=0%) and when adjusting for study duration (between-study variance=0.03), year of publication (between-study variance=0.02), mean age (between-study variance=0.02) or sex (between-study variance=0.03).

## DISCUSSION

We compared the efficacy and safety of cognitive enhancers regarding MMSE and AE outcomes to update our previous systematic review[6] and included studies with both aggregate data and IPD. Our results are in agreement with our previous systematic review,[6] and show that donepezil +memantine, donepezil alone and transdermal rivastigmine were the most effective treatments for improving MMSE score. However, heterogeneity was a major concern, which requires careful consideration before suggesting the use of cognitive enhancers, and particularly when the efficacy is not clear on the patient's characteristics. This was also captured by PIs, but their interpretation requires caution due to evidence of funnel plot asymmetry in the MMSE outcome. Overall, PIs are expected to include the true intervention effect expected in future studies, and they incorporate an extra component of variance, specifically between-study heterogeneity. In the absence of heterogeneity, CIs and PIs are equal. According to the P-score intervention ranking, both donepezil +memantine and transdermal rivastigmine had a favourable safety profile regarding AE, whereas the therapy with the least favourable profile was oral rivastigmine followed by donepezil. However, none of the estimated treatment effects were sufficiently precise when cognitive enhancers were compared with the placebo group. CINeMA suggested that within-study bias and reporting bias were the highest concerns for the MMSE outcome, whereas within-study bias and imprecision of effect estimates were the highest concerns for the AE outcome.

Overall, the choice among the different cognitive enhancers may depend on the patient's characteristics. In participants with moderate-to-severe cognitive impairment (defined by MMSE), a larger improvement in

cognitive performance was observed for donepezil and memantine, and their combination (donepezil +memantine), and these efficacy-related results are expected to also be reflected when a future study becomes available. The least effective cognitive enhancer in participants with moderate-to-severe cognitive impairment was oral rivastigmine. For patients with mild-to-moderate impairments based on MMSE scores, donepezil and transdermal rivastigmine were most likely the best performing cognitive enhancers. For patients with moderate-to-severe cognitive impairment, cognitive enhancers were well tolerated. For patients with mild-to-moderate cognitive impairment, all except for memantine and its combination with transdermal rivastigmine, were associated with increased odds of an AE, yet none of these results reached statistical significance. Overall, memantine was associated with lower odds of an AE than placebo, yet this was statistically significant only in the subnetwork analysis including aggregate data (ie, studies without IPD) and the meta-regression analysis using study duration as a covariate. However, acknowledging for heterogeneity in the network, PIs suggested that results are inconclusive and the odds of AE could not be differentiated between memantine and placebo. Of note, the accuracy of AE reporting may be impacted by the degree of cognitive impairment. Using IPD only and adjusting for MMSE baseline differences, (as shown in online supplemental appendix 16, MD: NMA of studies with IPD adjusted for baseline cognitive impairment), oral rivastigmine and galantamine improved MMSE score, whereas when adjusting for comorbidities only oral rivastigmine was effective, but results can change in a future study. Considering a MCID equal to 1.40 points,[21] the MDs of all cognitive enhancer regimens except for single-agent oral rivastigmine, galantamine and memantine, against placebo were clinically important for cognition, but these were associated with high uncertainty. However, the 1.40 MMSE cut-off value is not a widely adopted MCID. Also, high variability may be related to different populations included in the studies, such as genetic profiles, race and gender identity. Future studies should report this information to enable exploration of population characteristics that would benefit more, with a clinically important improvement, when using these treatments. Our results did not differ by participant characteristics sex, age and other medications, or by study characteristics, study duration and year of publication. However, these findings might be due to low power since meta-regression analyses depend on the number and size of studies, magnitude of the relationship between the covariate and effect size, along with its precision and heterogeneity.[25]

To the best of our knowledge, our study was the first to add IPD in an NMA of cognitive enhancers for patients with AD to produce treatment recommendations by patient characteristics. We followed the methods guidelines in the Cochrane Handbook for systematic reviews,[26] the reporting guidelines in the PRISMA-NMA and PRISMA-IPD statements[7 8] and evaluated credibility of findings using CINeMA.[15] Compared with previous systematic reviews, we included a larger number of studies and/or studies with shared IPD, compared in a wider range of cognitive enhancers.[6 27] Our results are in agreement with previous studies overall. Access to IPD allowed us to observe minor differences between the original published results and our reanalysis. An explanation in these differences may be that many studies used the last-observation-carried-forward imputation method, whereas we used the available case analysis when assessing MMSE. Another potential explanation might be that original studies excluded some patients, and hence used a smaller sample size.

Comparing NMA, results between aggregate data and IPD were in agreement. The only difference was observed in transdermal rivastigmine that was associated with a MCID of greater than 1.40 MMSE points against placebo in the aggregate data NMA compared with the IPD NMA, yet a statistically significant improvement was achieved in the IPD NMA. The inclusion of IPD in our NMA, allowed us to overcome potential reporting bias and to include IPD for (1) a study that we previously were unable to include since arm-level data were not reported in the RCT publication,[23] and (2) two studies that did not report MMSE results in their publications.[19 20] The use of IPD also allowed us to assess for potential effect modifiers that were not reported in the original publications (eg, comorbidities, additional medications) and explore for treatment-by-covariate interactions on the patient level. Several challenges were encountered during the IPD request from sponsors, showing that repositories are not a panacea (online supplemental appendix 21).

An important finding of our review is that the two-thirds of the published RCTs, were associated with high risk of bias for incomplete outcome data due to attrition, and the majority of these RCTs used the last-observation-carried-forward technique for missing data. This approach may bias results favouring cognitive enhancers, since the dropout rates were greater in the treatment group compared with the placebo group in 63% of the included studies and because dementia is a progressive disease. Of the 27 studies comparing treatment against placebo and reporting the number of dropouts, 17 studies had a greater dropout rate in the treatment group (treatment group: median dropout rate=28%, IQR (17%–39%); placebo group: median dropout rate=21%, IQR (15%–31%)). Last-observation-carried-forward is an inappropriate imputation method for AD studies, since it ignores expected deterioration of the patient's condition and stabilises the outcome at the value observed at the time of dropout (ie, the last observation).[28] Restricting to low risk of attrition bias studies, we found that galantamine was significantly associated with decreased odds of experiencing an AE.

Our study has limitations worth mentioning. First, we were unable to include IPD for all eligible studies (only 15% of the included RCTs shared their IPD), highlighting potential retrieval bias for IPD. However, recent

simulations have shown that combining IPD and aggregate data in an NMA can significantly improve precision, reduce bias and increase information compared with NMA relying on aggregated data alone.[29] Second, missing data are a big concern in the published RCTs for AD. We found high rates of dropouts from experiencing an AE and the patients' characteristics that may increase the chances of such adverse reactions prior to administering these cognitive enhancers should further be explored. To assess the impact of missing data in our NMA, we applied the informative missingness of difference in means.[30] However, future studies should explore the characteristics of missing participants and specific AEs. Third, the lack of studies in certain treatment comparisons may have affected the P-score calculation and treatment ranking. In particular, polytherapies were informed by maximum two studies, and ranking may have been in favour of the complex intervention group with the smaller number of studies.[31] For example, in MMSE the polytherapies including memantine in conjunction with one of the three treatments donepezil, galantamine, transdermal rivastigmine had a P-score ≥60%, but these all had wide 95% CIs for MD. As such, ranking should be interpreted with caution and along with the estimated effect sizes and their uncertainty measures. Fourth, the comparison-adjusted funnel plot for MMSE suggested there is an indication for small-study effects pointing to the treatment being better, and results should be interpreted with caution. This may also be related to the potential risk of funding bias, since the majority of the included studies were industry-sponsored and IPD were retrieved only from industry-sponsored studies favouring cognitive enhancers over placebo. Overall, MMSE score is only a surrogate maker for determining the impact of treatments on dementia. A full assessment that considers the potential impact of treatments on cognition, function and behavioural symptoms needs to be considered within the clinical context. Fifth, differences in patient characteristics, such as sex, were observed in the RCTs with provided IPD, which increased heterogeneity across studies. To account for these differences, we used the fully adjusted treatment effect estimates in the IPD analyses and the primary NMA analysis. Also, at the NMA level, we found that on average there were no important differences across treatment comparisons to threaten the transitivity assumption. Sixth, there are clinically important limitations associated with this review, including consistent definition of outcome measures across studies, a well-established MCID for the MMSE score, lack of consideration of drug doses due to inconsistent reporting and data retrieval bias that we were unable to overcome (15% of the studies shared their IPD). Future studies are needed to establish ranking efficacy in drug doses and combination of interventions across different disease severity categories. Seventh, the literature searches were conducted 5 years ago and additional relevant studies may be available. However, obtaining IPD in a timely manner was very challenging and required more time than anticipated (challenges to obtain IPD are outlined in online supplemental appendix 21). Similar to all systematic reviews, the evidence should be updated regularly.[32]

We expect that our findings will increase scientific knowledge, because people with AD require personalised medicine to optimise their healthcare. Well-conducted meta-analyses of IPD are considered the 'gold-standard' and influence patient care since patient-level data can be provided to facilitate tailored decision-making. However, results from meta-analyses of IPD are likely subject to retrieval bias and awareness of these limitations and their potential impact on findings is required (table 1).

**Author affiliations**
[1]Institute of Health Policy Management and Evaluation, University of Toronto, Toronto, Ontario, Canada
[2]Li Ka Shing Knowledge Institute, St. Michael's Hospital, Toronto, Ontario, Canada
[3]Department of Primary Education, University of Ioannina, Ioannina, Greece
[4]Centre for Reviews and Dissemination, University of York, York, UK
[5]Northern Ireland Hub for Trials Methodology Research, Queen's University Belfast, Belfast, UK
[6]Department of Biostatistics, University of Liverpool, Liverpool, UK
[7]Department of Medicine, University of Alberta, Edmonton, Alberta, Canada
[8]Department of Medicine, University of Calgary, Calgary, Alberta, Canada
[9]Department of Geriatric Medicine, University of Toronto, Toronto, Ontario, Canada
[10]Dalla Lana School of Public Health, University of Toronto, Toronto, Ontario, Canada

**Acknowledgements** We thank Laure Perrier for conducting the initial literature search, Alissa Epworth for updating it, and Becky Skidmore for peer reviewing the literature search. We thank Robert Peterson for his support on this study as a knowledge user. We would also like to thank Paul A Khan, Fatemeh Yazdi, Marco Ghassemi, John D Ivory, Charlene Soobiah, Erik Blondal, Joanne M Ho, Shirra Berliner and Carmen H Ng for screening some of the citations, data abstracting included studies or both. We thank Susan Le for helping with contacting the authors and sponsors of the included studies in this review. Finally, we thank Susan Le and Shazia Siddiqui for formatting the manuscript and assisting with the submission. We would also like to thank the following sponsors for sharing the data with us: 'This publication is based on research using data from data contributors, AbbVie, that has been made available through Vivli. Vivli has not contributed to or approved, and is not in any way responsible for, the contents of this publication.' 'This study, carried out under YODA Project #2017-1671, used data obtained from the Yale University Open Data Access Project, which has an agreement with Janssen Research & Development. The interpretation and reporting of research using this data are solely the responsibility of the authors and does not necessarily represent the official views of the Yale University Open Data Access Project or Janssen Research & Development.' This publication used data obtained from Eisai, GlaxoSmithKline and Novartis carried under www.ClinicalStudyDataRequest.com. This publication used data obtained from Lundbeck.

**Contributors** AAV, SES and ACT conceived and designed the study. AAV conducted the analyses, abstracted data, contacted sponsors, analysed data, interpreted results, appraised quality of results and wrote a draft manuscript. She is the guarantor of the present study. GS conducted the analyses, appraised quality of results and edited the manuscript. HMA coordinated the review, screened citations and full-text articles, abstracted data, appraised quality, cleaned the data, contacted sponsors and edited the manuscript. PR helped coordinate the study, screened citations and full-text articles, extracted and categorised data, appraised quality and edited the manuscript. SES and ACT interpreted results and edited the manuscript. ACT and HMA contacted authors. LS, MC, CT-S, DM, BRH and JH-L provided input into the design, interpreted results and edited the manuscript. All authors read and approved the final manuscript.

**Funding** This research was funded by the CIHR Drug Safety and Effectiveness Network (grant number 137713). The funder contributed to defining the scope of the systematic review but otherwise had no role in study design, data collection, analysis and interpretation of data. Data sharing sponsors were provided the ability to review at the time of submission of the manuscript for publication. AAV was previously funded by the Canadian Institutes of Health Research (CIHR) Banting

Postdoctoral Fellowship Program (No. 139157). GS and DM were funded from the European Union's Horizon 2020 (No. 754936). SES is funded by a Tier 1 Canada Research Chair in Knowledge Translation (No. N/A). ACT is funded by a Tier 2 Canada Research Chair in Knowledge Synthesis (No. N/A).

**Competing interests** None declared.

**Patient and public involvement** Patients and/or the public were not involved in the design, or conduct, or reporting, or dissemination plans of this research.

**Patient consent for publication** Not applicable.

**Provenance and peer review** Not commissioned; externally peer reviewed.

**Data availability statement** All data relevant to the study are included in the article or uploaded as supplementary information.

**ORCID iDs**
Areti Angeliki Veroniki http://orcid.org/0000-0001-6388-4825
Georgios Seitidis http://orcid.org/0000-0003-0856-1892
Mike Clarke http://orcid.org/0000-0002-2926-7257
Dimitris Mavridis http://orcid.org/0000-0003-1041-4592
Andrea C Tricco http://orcid.org/0000-0002-4114-8971

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
