## [Reviewer comments · BMJ Open]

ARTICLE DETAILS

TITLE (PROVISIONAL)	Comparative safety and efficacy of cognitive enhancers for Alzheimer's dementia: A systematic review with individual patient data network meta-analysis
AUTHORS	Veroniki, Areti; Ashoor, Huda; Rios, Patricia; Seitidis, Georgios; Stewart, Lesley; Clarke, Mike; Tudur-Smith, Catrin; Mavridis, Dimitris; Hemmelgarn, Brenda; Holroyd-Leduc, Jayna; Straus, Sharon; Tricco, Andrea

VERSION 1 – REVIEW

REVIEWER	Peter Watson University of Cambridge, MRC Cognition and Brain Sciences Unit
REVIEW RETURNED	28-Jun-2021

GENERAL COMMENTS	Page 13, line 264. On what measure were Donepezil and donepezil+memantine superior to placebo? I assume MMSE? Page 13, lines 262-279 and Page 15, lines 349-350. I think an important point is made here about what size of mean difference is clinically relevant. It appears that Donepezil+Memantine (lines 263-264) and Donepezil (line 278-279) in aggregate and IPD studies have mean differences with placebo which are large enough to suggest they have a clinically meaningful effect (differences above 1.40 - line 269) with Donepezil in studies using aggregate data only having a minimally overall clinical difference (lines 262-263). Why are just some of these differences cited here in the text e.g. galantamine+memantine in IPD studies is mentioned in line 278 but I don't see a mention of its mean difference? I do notice, however, that the sizes of the confidence intervals for the cited mean differences are of different sizes with the donepezil+memantine having a larger range of differences (0.07 to 5.07) than the Donepezil (0.051 to 2.32) which implies that the effectiveness of these treatments varies considerably depending on the individual which is acknowledged directly in the discussion (page 15, lines 349-350). Is there any reason, therefore, why some treatments may work better on some people than others? How would we know the optimal treatment or even combination of treatments to give to an individual? Is further work needed to establish this? How useful, therefore, is it to know how interventions differ from one another if their effectiveness varies so much depending upon the individual they are given to. Do we know what proportion or type of individuals show a clinically meaningful improvement compared to placebo using each of the interventions? Page 14, lines 306-316. Odds ratios are given here but I don't know how to interpret these as I don't know what a clinically meaningful
--

	magnitude of an odds ratio is. You give 1.40 as a clinically meaningful mean difference (page 13, line 269) so is there a similar one for the odds ratio? Page 14, lines 305-316 and page 16, lines 362-366. I am not clear from the confidence intervals that there is any difference in the odds of experiencing a SAE for oral rivastigmine, Donepezil and galantamine+memantine when these confidence intervals are so wide that they contain '1' suggesting that using a statistical test each one of these treatments would have found no difference in the likelihood of experiencing a SAE in either of these compared to a placebo which is acknowledged on line 312 which gives a confusing conclusion of "higher odds" (line 311) "...yet none of these comparisons were statistically significant" (lines 311-312) and in the discussion this statistical non-significance is further acknowledged (page 16, lines 362-366). I am also not sure how to interpret the p-scores and am not clear they are needed since I would conclude none of the above treatments differ from placebo due possibly to insufficiently precise estimates of the odds ratios of each treatment being obtained when compared with the placebo group. I noticed the text was written in a different typewriter type font (courier?) to the usual font.
--	---

REVIEWER	Jenny McCleery Oxford Health NHS Foundation Trust, Mental Health
REVIEW RETURNED	07-Jul-2021

GENERAL COMMENTS	Although I am not able to comment in detail on the statistics, as far as I can tell, this is a technically very well-conducted piece of work. The paper is well-written. The figures are clear and useful. My comments relate primarily to the clinical utility of the review and I hope are helpful to the authors. The protocol was registered appropriately, and the review has been written in line with appropriate methodological and reporting guidance. Introduction The introduction does not give a strong explanation of the need for an IPD network meta-analysis. There is a large body of work showing that all three cholinesterase inhibitors and memantine have at best benefits over placebo on cognition and global status which are small, within a narrow range, and of uncertain clinical importance. Therefore, it is not surprising that previous comparisons of the drugs with each other, including the authors' own earlier NMA, have demonstrated only differences which are fractions of probably clinically important differences. Conventional subgroup analyses in systematic reviews have been used to show that efficacy differences of these drugs from placebo vary to some extent with disease severity. The authors postulate that IPD data, which better characterize participants by (e.g.) disease severity and sex, may uncover groups of patients who show clearer differences in response to individual drugs. The authors should explain what specific, biologically-plausible hypotheses they were investigating, or whether their analysis was simply exploratory in this regard.
---

A reason given for the use of IPD is that published reports inconsistently report important covariates such as sex and disease severity. Disease severity as a potential effect modifier has been given particular prominence in the review, featuring in the abstract and discussion. The authors should explain how they intended to use IPD to make classification of disease severity more consistent. The reader needs to see what definitions were used for different severity categories. Appendices 16 and 17 refer to participants with “mild to moderate MMSE” and “moderate to severe MMSE” (this should be rephrased - presumably mild to moderate cognitive impairment, assessed with MMSE at baseline), but these categories are not defined.

Eligibility criteria

The search strategy appears very thorough, but the search date is >5 years old. It is unlikely that there is a substantial amount of more recent evidence, but when a search is this out-of-date, it would be good practice for the authors at least to run a search and identify any more recent eligible studies so that readers can see what quantity of data, if any, may be missing from the review.

Included studies are RCTs which assess MMSE and/or SAEs. On this basis, the authors include 80 RCTs, although not all contributed data to both outcomes. Their previous systematic review and NMA included 110 RCTs. It would be useful to know whether and how the large body of RCT evidence excluded from this review for both efficacy and safety outcomes differs from the included evidence.

The choice of outcome measures should be justified.

(a) MMSE is known to be highly education dependent and to lack sensitivity to change, especially at the milder end of the disease range. Further, there is no well-established MCID. The authors rely on an MCID of 1.4 which is an estimate derived from a single drug withdrawal trial in severe disease (DOMINO-AD). This cannot be taken as well-established, nor as applicable at all in mild or mild-to-moderate AD.

(b) Serious adverse events from these drugs are rare, but clinical decision-making is influenced very frequently by (less severe) AEs / tolerability. The authors should explain in the methods section what definition of SAE was used both to select studies and to extract data. From the definitions provided in Appendix 20 (note: text refers in error to Appendix 19) it appears that a mixture of data on SAEs as the term is usually understood (death, life-threatening events, hospitalisation, serious disability) and on any or all adverse events may have been used, which renders interpretation difficult.

IPD collection process

The authors have made a determined effort to obtain IPD data and it is disappointing that it was available for so few studies. This clearly limits the value added by IPD, but this is discussed.

Risk of bias and quality appraisal

This was done appropriately.

Synthesis

I am not able to comment in detail on the statistical methods.

	The authors appear to have pooled all doses of each drug. For clinical decision-making, distinctions between doses are important. Data on unlicensed drug doses have limited use clinically, but could have a significant effect on both efficacy and adverse event results. Were data included for participants receiving doses above and/or below the recommended dose range? The dose range should be included where study characteristics are described. Sensitivity analyses excluding doses outside the recommended therapeutic range should be considered. It would also be clinically useful to know whether the balance of efficacy and safety differs within the recommended dose range (e.g. donepezil 5mg or 10mg daily). Results The text of the review presents the main NMA results. If data are to be presented in the abstract on severity categories, then the relevant methods and data should be presented more accessibly in the text, along with the level of certainty and likely clinical importance of these results. There are some unexpected results presented in the appendices which do not agree with previous research, e.g. oral rivastigmine worsening cognition of placebo, memantine being associated with fewer SAEs than placebo, but these are not discussed, although they might tend to reduce the plausibility of the results as a whole. Can the authors comment on these? Discussion It is not clear from the objectives and methods of the review why disease severity has been selected from among other potential effect modifiers for particular emphasis in the discussion and in the abstract and this should be explained. It is of clinical interest, but was it a post hoc decision? There is a good discussion of the limitations of the included studies, but insufficient discussion of other, clinically important limitations of the review, including utility and consistency of definition of outcome measures, the absence of a well-established MCID for the MMSE, and lack of consideration of drug doses. As a result, I find it difficult to draw clinically applicable implications from this review of the data. The authors could consider ways to enhance clinical applicability, e.g. by ranking efficacy of defined doses of drugs for clearly defined disease severity categories and by being very explicit about how certain they are of any meaningful differences between the treatments.
--	--

VERSION 1 – AUTHOR RESPONSE

Reviewer 1 Comments:

1. I have some queries below on the usefulness of various methods used in this paper and the reporting of the results. I think the use of network analyses and ranking tools is not required and muddies the waters from the meta-analyses in an already overlong document and wonder about the interpretation of effect sizes claiming improvements in using interventions which do not show statistically significant differences between groups possibly down to the interventions having such variable effects on different people (as acknowledged in the abstract, page 5, line 39) and on groups

of patients who may differ on measures of confounding variables. It is, therefore, not clear which, if any, of these interventions would be of use on particular patients and what characteristics make some interventions more/less effective on some patients than others.

Authors' response: We understand that our paper is long and have included a lot of information in the supplementary material. However, this is customary with network meta-analyses (NMA), and for transparency we would prefer to report all the relevant results. Our primary aim for this paper was to examine the comparative efficacy and safety of cognitive enhancers by patient characteristics for managing Alzheimer's Dementia through an NMA. This is also described in our published protocol (<https://pubmed.ncbi.nlm.nih.gov/26769792/>), and this is also what we have agreed to do with our data providers. Besides, in a NMA we use the totality of evidence in a single model, and this allows us to increase certainty in the results compared to standard meta-analysis (e.g., donepezil vs. placebo NMA: 1.41 95% CI 0.51 to 2.32, meta-analysis: 1.65 95% CI 0.16 to 3.14). However, this may not be the case in the presence of considerable heterogeneity in the network.

Our interpretations are not solely based on statistical significance, but we also consider interpreting the whole 95% CI, the relevant minimal clinically important differences (MCID), presence of heterogeneity/inconsistency/small-study effects, number and size of studies, and characteristics of included studies (e.g., quality of studies). For example, a sparse network of trials may not have adequate power to detect a treatment effect.

We do not treat interpretation of treatment effects categorically, i.e., dichotomizing them to statistically significant/non-statistically significant treatment effects. We embrace uncertainty and consider how stable results can be in a future study using prediction intervals (PIs), which helps avoid overconfidence in the results. However, we emphasize our interpretation on the point estimate, since in a CI it is the most compatible value with the data. Therefore, we decided to discuss the point estimate even if a CI does not exclude the null value (e.g., MD=0). See also:

<https://www.nature.com/articles/d41586-019-00857-9>. For example, in the text we mention "Donepezil (MD= 1.41, 95% CI: 0.51 to 2.32) and donepezil+memantine (MD= 2.57, 95% CI: 0.07 to 5.07) were superior to placebo in terms of MMSE score (Appendices 16). PIs suggested results are not conclusive.", for statistically significant results, and "According to P-score, oral rivastigmine had the least favourable safety profile regarding SAE (OR= 1.26, 95% CI: 0.82 to 1.94, P-score= 16%), followed by donepezil (OR= 1.08, 95% CI: 0.87 to 1.35, P-score= 30%) and galantamine+memantine (OR= 1.03, 95% CI: 0.45 to 2.39, P-score= 43%), yet none of these comparisons were statistically significant different from placebo" for non-statistically significant results. Interpreting the point estimate, while acknowledging its uncertainty, can help avoid making false declarations of 'no effect' and resulting in overconfident conclusions.

2. I also am not sure how to interpret the meta-analyses looking at the same interventions on studies using either aggregate, IPD or both. For example If an intervention has a clinically meaningful overall difference on studies using both aggregate and IPD but not on studies using only IPD as appears to be the case with e.g. Donepezil which has a MD of 1,41 with IPD and aggregate (page 13, line 263) but only 0.70 for studies using IPD only (page 13, line 287) what do we conclude about the usefulness of this intervention on MMSE. Would you recommend using this intervention?

Authors' response: Our primary analysis was NMA of studies with both IPD and AD. As a subsequent analysis, we performed a sensitivity NMA of IPD only to explore the robustness of our results.

Variation in our findings is expected since our NMA of IPD+AD is based on 56 RCTs and 11,619 participants with between-study variance $\tau^2 = 5.75$, $I^2 = 96\%$ (donepezil for IPD+AD: MD= 1.41, 95% CI: 0.51 to 2.32), whereas our NMA of IPD only included 9 RCTs and 3,625 patients with $\tau^2 = 0.12$, $I^2 = 29\%$ (donepezil for IPD only: MD= 0.70, 95% CI: 0.01 to 1.40). Taking into consideration the treatment effect estimates along with their 95% CI, we can see that there is a considerable overlap between the CIs, but larger uncertainty is depicted in the IPD+AD estimate which is due to higher heterogeneity. Therefore, we conclude that the results of the two analyses are quite similar. It should be considered though that the network of the IPD only included newer and larger studies with smaller

treatment effects. This could explain a lower and more precise MD in the network of studies with IPD only.

3. Page 9, lines 147-152 and line 162. What software was used to fit the funnel plots and perform the random effects meta-analysis. I assume that there was considerable heterogeneity between studies (e.g. large I^2 s) to justify using a random effects meta analysis?

Authors' response: We describe all relevant details for our analysis in Appendix 1. We included most of this information in the supplementary file, since our paper was already very long and to make the paper easier to follow. In particular, we report:

"We assessed reporting bias based on the comparison-adjusted funnel plot since there are no established statistical methods to explore reporting bias. We used the netfunnel command in Stata to produce the comparison-adjusted funnel plot.4 [...] Meta-analysis and NMA at the 2nd stage were conducted in the RStudio using R version 3.6.2 and the meta19 and netmeta20 packages, respectively."

We disagree with the selection of the meta-analysis model based on considerable heterogeneity; the random-effects model is not a remedy for heterogeneity. As highlighted in the Cochrane Handbook, the choice of the model should be decided a priori. The choice of the model should not be based on heterogeneity tests or measures (as they have different properties in different cases), but only on prior beliefs. We expected that the eligible studies would vary according to different study designs and effect modifiers, and hence at the protocol stage we decided that the random-effects model would be the most appropriate to use.

4. Page 69, Appendix 12, The left funnel plot appears to have several studies outside the top right edge of the funnel plot. Does this imply bias? Shouldn't the studies in the plot be symmetric? Did you use trim and fill methods to adjust for any bias as obtained from the funnel plots. I can't see any mention of trim and fill in the paper.

Authors' response: The MMSE comparison-adjusted funnel indicated the presence of small-study effects. This is also discussed in the text.

Risk of bias and IPD integrity, lines 197-198: "Comparison-adjusted funnel plots suggested there is indication for small-study effects (see Appendix 12)."

Discussion section, lines 424-426: "Fourth, the comparison-adjusted funnel plot for MMSE suggested there is an indication for small-study effects pointing to the treatment being better, and results should be interpreted with caution."

We agree with the reviewer that this can be explored further, and hence we performed the Copas selection bias model to explore publication bias. We do not agree with the use of the trim and fill method, since it is not optimal (<https://pubmed.ncbi.nlm.nih.gov/19836925/>). As also explained in the Cochrane Handbook, the trim and fill method builds on the assumption of a symmetric funnel plot with adjusted intervention effects designed to match what would have been observed in the absence of publication bias. However, the method does not take into account reasons for funnel plot asymmetry other than publication bias, and 'corrected' intervention effect estimates should be interpreted with great caution.

We explored the treatment comparison donepezil vs. placebo, in which we observed asymmetry and it was informed by more than 10 studies. We used the Copas selection model to explore publication bias, as described in lines 124-131:

“When at least 10 studies were available for each treatment against placebo, publication bias and small-study effects were examined visually using the comparison adjusted funnel plot under the fixed-effect model.³ When a funnel plot asymmetry was detected, we performed the Copas selection for the treatment comparisons that were informed by at least 10 studies and for which asymmetry was evident in the funnel plot. We explored the possibility that this was due to publication bias,¹⁴ and made moderate assumptions about the probability of publication of the smaller and larger (in terms of standard error) studies. We assumed that the smallest study had a probability of publication equal to 40-50% and the largest study had a probability of 80-90%.”

Lines 197-202: “Comparison-adjusted funnel plot for MMSE suggested there is indication for small-study effects (see Appendix 12). In contrast to the standard meta-analysis (MD=1.65 95% CI (0.16, 3.14)), the Copas selection model estimated a pooled treatment effect for donepezil vs. placebo MD=1.87 95% CI (1.55, 2.20) with between-study variance $\tau^2= 1.95$, and correlation coefficient -0.45 (-0.76, -0.01) reflecting the belief that the propensity for publication was associated with the observed effect size”

5. Page 10, lines 168-170. There are a lot of figures and diagrams and 104 pages including appendices in this submission. Are these all necessary? For example do you need to use both P-scores and surface under cumulative ranking curves (SUCRAs) to rank the interventions or indeed either of these? I would like to see more motivation given for their use or these methods dropped from the paper. To be honest I have not seen P-scores or SUCRAs used in any meta-analysis publication before because the interest is in the pooled sizes of the difference or odds ratio between pairs of interventions (e.g. forest plots) and bias (e.g. funnel plots and Eggers test) and also trim and fill methods. Looking at the relative sizes of these pooled differences or odds ratios one could come to conclusions about the relative effectiveness of the treatments without the need to use further pictorial or numerical methods.

Authors' response: Reporting ranking measures, such as P-scores and SUCRAs is common practice in systematic reviews with NMAs and help with interpretation of results in particular when a big number of interventions and studies is explored in NMA, see also the PRISMA extension to NMA (<http://www.prisma-statement.org/Extensions/NetworkMetaAnalysis>). Ranking measures are only complementary and are interpreted along with the relevant effect estimates and their confidence intervals. A presentation of a single treatment ranking statistic is not informative. As clarified in the text, we present P-scores in all analyses apart from the network meta-regression analyses, since all analyses apart from the NMA meta-regression were conducted in a frequentist setting. P-scores are the frequentist equivalent ranking measure of SUCRAs (see <https://pubmed.ncbi.nlm.nih.gov/26227148/>).

Lines 144-146: “We ranked the interventions for each outcome using the P-scores (and SUCRAs [surface under the cumulative ranking curve] in meta-regression analysis), and present them in a rank-heat plot”

According to our response in comment #1 long supplementary files and results of multiple analyses are customary with NMA, and as per PRISMA-NMA we would prefer to be transparent and report all the relevant results. Our study design was a systematic review with an IPD-NMA, as also described in our published protocol, in order to compare all available treatments for this clinical setting. We have also added a couple of sentences (see underlined) in the background, as shown in the paragraph below.

Lines 74-84: “In AD, disease severity and sex are potential effect modifiers. However, aggregate data and covariates of interest (e.g., sex, disease severity) are not consistently reported across randomized clinical trials (RCTs).⁶ The use of IPD has several advantages, such as it allows for the exploration of the relationship between treatment effects and patient-level characteristics, and it overcomes restrictions in using the information reported in the publication among others. The aim of this study was to examine the comparative efficacy and safety of cognitive enhancers for patients with

different characteristics, such as severities of AD and for females versus males through a systematic review and individual patient data (IPD) NMA. NMA is an extension of standard meta-analysis synthesizing different sources of evidence from a network of RCTs comparing different treatments within a single model. NMA can provide treatment effect estimates for treatment comparisons that have not studied in a head-to-head study.”

6. Page 11, line 208. By not balanced do you mean that the groups differed on measures of interest or were of different sizes? If the former on what measures did these groups differ and what is the consequence of these differences in interpreting the meta-analyses. Have any differences found between groups on confounder variables been adjusted for in the meta-analyses and if not, could any differences found between the groups be simply due to the different levels of confounder variables present within these groups and not to the interventions themselves?

Authors’ response: We clarified the text as shown below:

Lines 176-180: Important patient characteristics, such as percent of male and dropout rates, were not balanced across groups in the RCTs with provided IPD (Appendix 6). Comparing study and patient characteristics of available and non-available IPD when a study was industry-sponsored, we found differences in the year of study publication, study size, and absolute mean difference (Appendix 7).

We also comment on the consequence of these differences in the discussion section:

Lines 429-434: “Fifth, differences in patient characteristics, such as sex, were observed in the RCTs with provided IPD, which increased heterogeneity across studies. To account for these differences, we used the fully adjusted treatment effect estimates in the IPD analyses and the primary NMA analysis. Also, at the NMA level, we found that on average there were no important differences across treatment comparisons to threaten the transitivity assumption.”

7. Page 11, lines 224-225. It states here that there is a high potential risk of bias in 2/3rds of the studies in the meta-analysis. Does this bias carry through to your meta-analyses? Is your meta-analysis only as unbiased as the quality including bias (or not) of the studies in it?

Authors’ response: We agree that the quality of studies can importantly impact the quality of the meta-analytical results, and this is also highlighted in the assessment of the credibility in the NMA findings using Confidence in Network Meta-Analysis (CINeMA). However, as explained in Appendix 1:

“We applied an available case analysis for each study, since we were unable to install R packages in most sponsor-specific platforms, and hence we applied a consistent approach across all IPD datasets. We explored the impact of missing data during the second stage of analysis. Reasons for missing participants and time taken to have a serious adverse event were captured (when available).”

To explore the impact of the high risk of bias for missing data in NMA, we performed the ‘informative missingness difference of means’ (IMDoM) imputation method and a sensitivity analysis restricting to low risk of attrition bias studies; results are presented in Appendix 16. Also, this limitation was considered in the discussion section:

Lines 397-417: “An important finding of our review is that the two thirds of the published RCTs, were associated with high risk of bias for incomplete outcome data due to attrition, and the majority of these RCTs used the last-observation-carried-forward technique for missing data [...] Last-observation-carried-forward is an inappropriate imputation method for Alzheimer’s Dementia studies, since it ignores expected deterioration of the patient’s condition and stabilizes the outcome at the value observed at the time of dropout (i.e., the last observation).²⁴ Restricting to low risk of attrition bias studies, we found that galantamine was significantly associated with decreased odds of experiencing a SAE[...] Second [limitation], missing data is a big concern in the published RCTs for Alzheimer’s Dementia. To assess the impact of missing data in our NMA, we applied the informative missingness of difference in means.²⁶ ”

8. Page 16, lines 368-372. There is a mention here of possible reasons for imprecision in the measurement of SAE and mention of adjusting for MMSE score. I am not sure to which meta-analyses the adjustment for MMSE mentioned here refers to since the results where MMSE is mentioned on page 13 don't appear to mention any adjustment for any measure of MMSE but instead seek to to establish if MMSE scores differ between interventions?

Authors' response: We present the results of our primary analyses in the main manuscript, and results of subsequent analyses in the supplementary to make the manuscript more manageable and easier to read. In lines 209-210, we mention "Additional analyses are presented in Appendices 15-16" We also moved information on our subsequent analysis from previous Appendix 16 to the main manuscript.

Our subsequent analyses with covariate adjustments include 1) NMA analysis with IPD only and adjusted at the 1st stage of analysis (i.e., when reducing IPD to aggregate data) for any of the following variables that were available in each study: age, sex, severity of Alzheimer's disease (e.g., baseline Mini-Mental State Examination [MMSE] level), presence of behavioural disturbance, comorbidity, and other medications, 2) NMA meta-regression of aggregate data, and 3) subgroup/sensitivity NMA analyses. All results are presented for each intervention comparison separately to explore potential effect modifiers in the network.

In any case, we clarified the sentence in the discussion section as (lines 360-364):

"Using IPD only and adjusting for MMSE baseline differences (as shown in Appendix 16), Mean Difference: NMA of studies with IPD adjusted for baseline cognitive impairment), oral rivastigmine and galantamine improved MMSE score, whereas when adjusting for comorbidities only oral rivastigmine was effective, but results can change in a future study."

9. Page 16, lines 371-372. "...results can change in a future study." This does not inspire confidence and makes me wonder just how robust the results are in this paper. Why would future analyses yield different results - is this to do with the replicability problem or in carrying out group comparisons adjusting for confounders or on different populations?

Authors' response: This interpretation is based on the derived prediction intervals (PIs). Given the high heterogeneity across studies, which was a major concern in the NMA results, as also shown with CINeMA, our results may change when a future study becomes available, and this was also captured by PIs. As described in the Cochrane Handbook

(<https://training.cochrane.org/handbook/current/chapter-10>), PIs are helpful for presenting the extent of between-study variation, and particularly when we obtain short CIs around the pooled random-effects estimate in the presence of high heterogeneity, which is also our case. In particular, PIs show an estimate of an interval in which the treatment effect will fall, with a certain probability, in a future individual setting, given what has already been observed.

10. Page 12, Similarly I would like to see a mention of what the network meta-analysis is adding to the forest and funnel plots used to obtain effect sizes and confidence intervals and bias tests in order to justify its use. What do network meta-analyses add to the results from the forest and funnel plots which appear to constitute the bulk of the results from the meta-analyses on pages 13 to 15. This is such a method heavy paper that a more streamlined paper would be clearer.

Authors' response: Our initial aim was to assess safety and efficacy of cognitive enhancers for Alzheimer's dementia using the totality of evidence in an NMA (please see also responses to comments #1 and #5). We understand that a NMA is an advanced method, but it is a standard approach used in health technology assessments (HTA) for regulatory approval. NMAs are best designed for conditions with multiple interventions, many combinations of direct or indirect

interactions, to answer more relevant and broader clinical questions, obtain treatment effect estimates for an entire network instead of scanning each individual pairwise comparison, to give the 'full picture' to clinicians, guideline developer, policy makers, and patients, to gain precision by considering all available evidence in a single model, and to provide a 'ranking' hierarchy of all available interventions using summary results.

11. Page 13, line 264. On what measure were Donepezil and donepezil+memantine superior to placebo? I assume MMSE?

Authors' response: Thank you for this comment. We have clarified this in the text as shown below (lines 223-225):

"Donepezil (MD= 1.41, 95% CI: 0.51 to 2.32) and donepezil+memantine (MD= 2.57, 95% CI: 0.07 to 5.07) were superior to placebo in terms of MMSE score (Appendix 15)."

12. Page 13, lines 262-279 and Page 15, lines 349-350. I think an important point is made here about what size of mean difference is clinically relevant. It appears that Donepezil+Memantine (lines 263-264) and Donepezil (line 278-279) in aggregate and IPD studies have mean differences with placebo which are large enough to suggest they have a clinically meaningful effect (differences above 1.40 - line 269) with Donepezil in studies using aggregate data only having a minimally overall clinical difference (lines 262-263). Why are just some of these differences cited here in the text e.g. galantamine+memantine in IPD studies is mentioned in line 278 but I don't see a mention of its mean difference? I do notice, however, that the sizes of the confidence intervals for the cited mean differences are of different sizes with the donepezil+memantine having a larger range of differences (0.07 to 5.07) than the Donepezil (0.051 to 2.32) which implies that the effectiveness of these treatments varies considerably depending on the individual which is acknowledged directly in the discussion (page 15, lines 349-350). Is there any reason, therefore, why some treatments may work better on some people than others? How would we know the optimal treatment or even combination of treatments to give to an individual? Is further work needed to establish this? How useful, therefore, is it to know how interventions differ from one another if their effectiveness varies so much depending upon the individual they are given to. Do we know what proportion or type of individuals show a clinically meaningful improvement compared to placebo using each of the interventions?

Authors' response: Thank you for this remark. We added the results in the text, where missing.

Lines 223-229: "Studies in this NMA compared all available treatments. Donepezil (MD= 1.41, 95% CI: 0.51 to 2.32) and donepezil+memantine (MD= 2.57, 95% CI: 0.07 to 5.07) were superior to placebo in terms of MMSE score [...] Transdermal rivastigmine (MD= 2.11, 95% CI: -0.04 to 4.26), and the combinations donepezil+memantine, galantamine+memantine (MD= 2.24, 95% CI: -2.13 to 6.61), and transdermal rivastigmine+memantine (MD= 1.79, 95% CI: -1.70 to 5.27) were associated with a minimal clinically important difference (MCID; above 1.40)".

Lines 239-241: "The MCID results were in agreement with the NMA of IPD and aggregate data, and donepezil+memantine (MD= 2.71, 95% CI: -0.17 to 5.60) was likely the most effective in improving MMSE score (P-score= 76%)."

Our results represent the average mean difference between an intervention and placebo for each treatment comparison in the network. Combining the effect sizes from all the eligible studies in a NMA, we can only infer on the average effect and its clinical significance. Using the IPD we could explore the proportion of participants with a clinically meaningful improvement compared to placebo across all interventions. However, we did not initially capture this information, and currently we do not have access to the IPD for further exploration, since access to the IPD was only for a certain amount of time. We agree with the reviewer that treatment effectiveness may vary depending on the individual's characteristics, and this is what we explored through our subsequent analyses. We discussed this in the network meta-analysis section lines 250-278 and 312-327 and the discussion section lines 343-354:

“Overall, the choice among the different cognitive enhancers may depend on the patient’s characteristics. In participants with moderate to severe cognitive impairment (defined by MMSE), a larger improvement in cognitive performance was observed for donepezil and memantine, and their combination (donepezil+memantine), and these efficacy-related results are expected to also be reflected when a future study becomes available. The least effective cognitive enhancer in participants with moderate to severe cognitive impairment was oral rivastigmine. For patients with mild to moderate impairments based on MMSE scores, donepezil and transdermal rivastigmine were most likely the best performing cognitive enhancers. For patients with moderate to severe cognitive impairment, cognitive enhancers were well tolerated. For patients with mild to moderate cognitive impairment, all except for memantine and its combination with transdermal rivastigmine, were associated with increased odds of a SAE, yet none of these results reached statistical significance.”

We also point out the following in the discussion section:

Lines 434-439: “Sixth, there are clinically important limitations associated with this review, including consistent definition of outcome measures across studies, a well-established MCID for the MMSE score, lack of consideration of drug doses due to inconsistent reporting and data availability bias that we were unable to overcome (15% of the studies shared their IPD). Future studies are needed to establish ranking efficacy in drug doses and combination of interventions across different disease severity categories.”

13. Page 14, lines 306-316. Odds ratios are give here but I don't know how to interpret these as I don't know what a clinically meaningful magnitude of an odds ratio is. You give 1.40 as a clinically meaningful mean difference (page 13, line 269) so is there a similar one for the odds ratio?

Authors’ response: As already reported in Appendix 1, we considered an OR=1 as a clinically relevant value:

“For imprecision, we considered a MD=1.4 and a OR=1 as a clinically important size of effect for MMSE and SAE, respectively, and followed the CINeMA guidelines for exploring whether statistical significance and clinical importance coincide.”

14. Page 14, lines 305-316 and page 16, lines 362-366. I am not clear from the confidence intervals that there is any difference in the odds of experiencing a SAE for oral rivastigmine, Donepezil and galantamine+memantine when these confidence intervals are so wide that they contain '1' suggesting that using a statistical test each one of these treatments would have found no difference in the likelihood of experiencing a SAE in either of these compared to a placebo which is acknowledged on line 312 which gives a confusing conclusion of "higher odds" (line 311) "...yet none of these comparisons were statistically significant" (lines 311-312) and in the discussion this statistical non-significance is further acknowledged (page 16, lines 362-366). I am also not sure how to interpret the p-scores and am not clear they are needed since I would conclude none of the above treatments differ from placebo due possibly to insufficiently precise estimates of the odds ratios of each treatment being obtained when compared with the placebo group.

Authors’ response: Thank you for this comment. We have updated our discussion section clarifying that our conclusion is based on the P-score ranking and we have also emphasized that treatment effect estimates are not sufficiently precise (see underlined). This is also suggested in our assessment with the CINeMA tool:

Lines 335-341: “According to the P-score intervention ranking, both donepezil+memantine and transdermal rivastigmine had a favourable safety profile regarding SAE, whereas the therapy with the least favourable profile was oral rivastigmine followed by donepezil. However, none of the estimated treatment effects were sufficiently precise when cognitive enhancers were compared with the placebo

group. CINeMA suggested that within-study bias and reporting bias were the highest concerns for the MMSE outcome, whereas within-study bias and imprecision of effect estimates were the highest concerns for the SAE outcome.”

Similarly we updated the results section:

Lines 23-27: “According to P-score, oral rivastigmine had the least favourable safety profile regarding SAE (OR= 1.26, 95% CI: 0.82 to 1.94, P-score= 16%), followed by donepezil (OR= 1.08, 95% CI: 0.87 to 1.35, P-score= 30%) and galantamine+memantine (OR= 1.03, 95% CI: 0.45 to 2.39, P-score= 43%), yet none of these comparisons were statistically significant different from placebo”

15. I noticed the text was written in a different typewriter type font (courier?) to the usual font.

Authors’ response: Thank you. We also have noticed this, which was surprising, but this is most probably due to the journal’s online system.

Reviewer 2 Comments:

1. Although I am not able to comment in detail on the statistics, as far as I can tell, this is a technically very well-conducted piece of work. The paper is well-written. The figures are clear and useful. My comments relate primarily to the clinical utility of the review and I hope are helpful to the authors.

Authors’ response: Thank you for your helpful comments.

2. The protocol was registered appropriately, and the review has been written in line with appropriate methodological and reporting guidance.

Authors’ response: Thank you.

3. The introduction does not give a strong explanation of the need for an IPD network meta-analysis. There is a large body of work showing that all three cholinesterase inhibitors and memantine have at best benefits over placebo on cognition and global status which are small, within a narrow range, and of uncertain clinical importance. Therefore, it is not surprising that previous comparisons of the drugs with each other, including the authors’ own earlier NMA, have demonstrated only differences which are fractions of probably clinically important differences.

Authors’ response: We decided to keep our introduction as short as possible, so as to include more details in the remaining sections of the manuscript, since most of this information is reported in our published protocol (<https://pubmed.ncbi.nlm.nih.gov/26769792/>). To show the need of IPD we have added the following text:

Lines 76-78: “The use of IPD has several advantages, such as it allows for the exploration of the relationship between treatment effects and patient-level characteristics, and it overcomes restrictions in using the information reported in the publication among others.”

We have also updated the background as (see underlined):

Lines 66-72: “Pharmacological treatment for AD predominantly consists of cholinesterase inhibitors (donepezil, galantamine, rivastigmine) and the N-methyl-d-aspartate (NMDA) receptor antagonist, memantine. All three cholinesterase inhibitors and memantine are currently the only effective licensed treatments for dementia,³ but their clinical effect can be small and there is no convincing evidence that they modify the disease process in AD.⁴ Also, it is unclear whether galantamine, rivastigmine, or donepezil should be used by patients with severe AD, or whether memantine is the optimal treatment for severe AD.⁵”

4. Conventional subgroup analyses in systematic reviews have been used to show that efficacy differences of these drugs from placebo vary to some extent with disease severity. The authors postulate that IPD data, which better characterize participants by (e.g.) disease severity and sex, may uncover groups of patients who show clearer differences in response to individual drugs. The authors should explain what specific, biologically-plausible hypotheses they were investigating, or whether their analysis was simply exploratory in this regard.

Authors' response: Our intention was to explore statistical heterogeneity in terms of study and patient level variables. The use of IPD allowed us to explore potential effect modifiers that we were unable to do so using aggregate data from published studies in our previous NMA (<https://pubmed.ncbi.nlm.nih.gov/29131306/>). We were able to adjust study data with available IPD (i.e., when reducing IPD to aggregate data at 1st stage) for any of the following variables that were available in each study: age, sex, severity of Alzheimer's disease (e.g., baseline Mini-Mental State Examination [MMSE] level), presence of behavioural disturbance, comorbidity, and other medications. Hence, at 2nd stage we combined adjusted estimates with the remaining aggregate data in a network meta-analysis, which helped us slightly reduce heterogeneity across studies. However, we were unable to combine all IPD in a one-stage meta-analysis model, since we were provided access to IPD through different online platforms.

As also explained in the discussion section:

Lines 388-394: "The inclusion of IPD in our NMA allowed us to overcome potential reporting bias and to include IPD for 1) a study that we previously were unable to include since arm-level data were not reported in the RCT publication,²⁰ and 2) two studies that did not report MMSE results in their publications.^{17,18} The use of IPD also allowed us to assess for potential effect modifiers that were not reported in the original publications (e.g., comorbidities, additional medications) and explore for treatment-by-covariate interactions on the patient-level."

Also, in the NMA results section:

Lines 218-219: "Two studies^{17,18} did not report MMSE in the final publication, but in the retrieved IPD we were able to use data for this outcome."

5. A reason given for the use of IPD is that published reports inconsistently report important covariates such as sex and disease severity. Disease severity as a potential effect modifier has been given particular prominence in the review, featuring in the abstract and discussion. The authors should explain how they intended to use IPD to make classification of disease severity more consistent. The reader needs to see what definitions were used for different severity categories. Appendices 16 and 17 refer to participants with "mild to moderate MMSE" and "moderate to severe MMSE" (this should be rephrased - presumably mild to moderate cognitive impairment, assessed with MMSE at baseline), but these categories are not defined.

Authors' response: We accounted for disease severity in the analysis of studies with available IPD by adjusting for the observed patient-specific baseline MMSE at the 1st stage. Indeed, AD severity was not reported in 8 publications of the 56 RCTs in the MMSE outcome. Of these, one study had available IPD. We combined IPD adjusted for AD severity in a NMA, and results are presented in Appendix 16 'Mean Difference: NMA of studies with IPD adjusted for cognitive impairment, assessed with MMSE at baseline'. In a separate analysis we combined studies with aggregate data only in a network meta-regression accounting for AD severity as classified by the original publications. These results are presented in Appendix 16, 'Mean Difference: Studies with Mild to Moderate cognitive impairment, assessed with MMSE at baseline' and 'Mean Difference: Studies with Moderate to Severe cognitive impairment, assessed with MMSE at baseline' (note that this was rephrased as suggested by the reviewer). Hence, we did not proceed to any further classifications for the studies with available IPD. We categorized aggregate data according to MMSE scores using the National

Institute for Health and Care Excellence categories: mild (21–24), moderate (10–20), severe (<10), and we clarify this in the text.

This is also presented in Appendix 1 (see clarification in underlined sentence) “Additional NMA analyses included: 1) subgroup analysis for industry vs. publicly sponsored studies, for studies with available IPD vs. studies with aggregate data (unadjusted estimates), and for AD severity, classified according to MMSE scores using the National Institute for Health and Care Excellence categories: mild (21–24), moderate (10–20), severe (<10), 2) network meta-regression accounting for study duration, year of publication, mean age, and sex (% of male participants) effect modifiers separately and assuming a common regression coefficient across comparisons (studies with aggregate data were used only; studies with available IPD were pooled in a NMA separately adjusted for available covariates at first stage)”

6. The search strategy appears very thorough, but the search date is >5 years old. It is unlikely that there is a substantial amount of more recent evidence, but when a search is this out-of-date, it would be good practice for the authors at least to run a search and identify any more recent eligible studies so that readers can see what quantity of data, if any, may be missing from the review.

Authors' response: We understand that the literature search of this review is outdated. However, obtaining the IPD was very challenging and required a lot more time than anticipated. We outline the challenges we encountered during the IPD request from sponsors in Appendix 21. Given that the time to obtain IPD was >1 year after a sponsor's positive response, updating the literature search and following the same process would require an important amount of time plus funding. This research was funded by the CIHR Drug Safety and Effectiveness Network, and currently funding cannot be extended for another update. Therefore, it is not feasible to update the present review.

We highlight this limitation in our discussion section:

Lines 439-443: “Seventh, the literature searches were conducted 5 years ago and additional relevant studies may be available. However, obtaining IPD in a timely manner was very challenging and required more time than anticipated (challenges to obtain IPD are outlined in Appendix 21). Similar to all systematic reviews, the evidence should be regularly updated.”

7. Included studies are RCTs which assess MMSE and/or SAEs. On this basis, the authors include 80 RCTs, although not all contributed data to both outcomes. Their previous systematic review and NMA included 110 RCTs. It would be useful to know whether and how the large body of RCT evidence excluded from this review for both efficacy and safety outcomes differs from the included evidence.

Authors' response: Our previous systematic review included different study designs, i.e., RCTs, quasi-RCTs, and nonrandomized studies, which reported on multiple outcomes, including cognition, function, behaviour, global status, behaviour, SAE, falls, diarrhea, bradycardia, vomiting, headache, nausea. In the present review we included RCTs only and considered only the ones assessing the MMSE and SAE outcomes. In particular, we included 56 RCTs in the MMSE, but these contributed more patients (11,619) compared to the previous NMA (10,446 participants), which is due to the available IPD. In the studies with shared IPD, we included all participants with available data as originally randomized. In the SAE outcome we included 45 RCTs (15,649 participants) compared to the previous review where we included 48 studies (14,189 participants). This is because in the present review we restricted to published and English RCTs only which is due to excluding unpublished and non-English RCTs. We clarified this in the text:

Lines 101-103: “We included published and English RCTs that assessed cognition via the Mini-Mental State Examination (MMSE; efficacy and primary outcome) and/or serious adverse events (SAE; safety outcome) in adults with Alzheimer’s dementia”

8. The choice of outcome measures should be justified:

(a) MMSE is known to be highly education dependent and to lack sensitivity to change, especially at the milder end of the disease range. Further, there is no well-established MCID. The authors rely on an MCID of 1.4 which is an estimate derived from a single drug withdrawal trial in severe disease (DOMINO-AD). This cannot be taken as well-established, nor as applicable at all in mild or mild-to-moderate AD.

(b) Serious adverse events from these drugs are rare, but clinical decision-making is influenced very frequently by (less severe) AEs / tolerability. The authors should explain in the methods section what definition of SAE was used both to select studies and to extract data. From the definitions provided in Appendix 20 (note: text refers in error to Appendix 19) it appears that a mixture of data on SAEs as the term is usually understood (death, life-threatening events, hospitalisation, serious disability) and on any or all adverse events may have been used, which renders interpretation difficult.

Authors’ response: Although MMSE is not a perfect outcome, it is the most commonly used measure of cognition for Alzheimer’s Dementia. A recent survey among knowledge users from three provinces in Canada regarding the prioritization of outcomes and measures in a systematic review showed that the most relevant outcome for patients, caregivers, policymakers, and geriatricians, regarding cognition, was the MMSE followed by the Clock Drawing Test and the Clinical Dementia Rating scale (<https://pubmed.ncbi.nlm.nih.gov/30771447/>).

We also considered including the SAE outcome since many decision-makers need this information and choose this as an important outcome. Regarding the SAE definition we followed the definition as reported in the individual studies. As noted by the reviewer, we provide study definitions in Appendix 19. Appendix number has been corrected in the text, thank you for pointing this out.

9. The authors have made a determined effort to obtain IPD data and it is disappointing that it was available for so few studies. This clearly limits the value added by IPD, but this is discussed.

Authors’ response: This is indeed disappointing given all the effort, time, and funding spent over many years. We contacted authors and sponsors of the eligible RCTs to retrieve IPD, and in cases that multiple sponsors were reported, we contacted all of them. To facilitate IPD retrieval, we also contacted the Clinical Study Data Request and Yale University Open Data Access data sharing platforms. Failure to retrieve all available RCTs and restricting analysis to studies with available IPD can severely impact NMA findings and decision-making.

To overcome potential retrieval bias, medical journals endorse standards for reporting of study results, such as the Consolidated Standards of Reporting Trials (CONSORT) checklist (<http://www.consort-statement.org/>). Despite these efforts, study data are inconsistently reported, and missing evidence is a substantial problem, and one of the greatest threats to the validity of results from a systematic review and meta-analysis.

10. Risk of bias and quality appraisal: This was done appropriately.

Authors’ response: Thank you.

11. Synthesis: I am not able to comment in detail on the statistical methods.

The authors appear to have pooled all doses of each drug. For clinical decision-making, distinctions between doses are important. Data on unlicensed drug doses have limited use clinically, but could have a significant effect on both efficacy and adverse event results. Were data included for participants receiving doses above and/or below the recommended dose range? The dose range

should be included where study characteristics are described. Sensitivity analyses excluding doses outside the recommended therapeutic range should be considered. It would also be clinically useful to know whether the balance of efficacy and safety differs within the recommended dose range (e.g. donepezil 5mg or 10mg daily).

Authors' response: Thank you for pointing this out. We have included the dose range used across studies in the characteristics table Appendix 4. For studies that examined different dosages of the same medication in different treatment arms, dosages selected for analysis were consistent with those approved for use in Canada. We clarified this in Appendix 4. However, we were not able to examine dosages in our NMA because many of the studies reported only a dosage range, and others did not report dose information in the publication. Overall, the dosages included in the analysis were:

- Donepezil 5-10 mg o.d.
- Galantamine 16-24 mg o.d.
- Memantine 5-10 mg b.i.d.
- Rivastigmine 3-6 mg b.i.d.
- Rivastigmine transdermal 9.5-13.3 mg o.d.

12. The text of the review presents the main NMA results. If data are to be presented in the abstract on severity categories, then the relevant methods and data should be presented more accessibly in the text, along with the level of certainty and likely clinical importance of these results.

Authors' response: We moved information from Appendix 16 relevant to the additional analyses using IPD and aggregate data to the main manuscript.

13. There are some unexpected results presented in the appendices which do not agree with previous research, e.g. oral rivastigmine worsening cognition cf placebo, memantine being associated with fewer SAEs than placebo, but these are not discussed, although they might tend to reduce the plausibility of the results as a whole. Can the authors comment on these?

Authors' response: Our NMA results suggest that all cognitive enhancers are more effective than placebo, but only donepezil and donepezil+memantine are statistically significantly better in terms of MMSE. Also, donepezil+memantine is most likely the most effective cognitive enhancer in improving MMSE score (see Appendix 15). Memantine was associated with lower odds of a SAE than placebo, yet this was statistically significant only in the sensitivity NMA of studies with remaining aggregate data (i.e., studies with non-available IPD) and the meta-regression analysis using study duration as a covariate. Our primary analysis including both IPD and aggregate data was associated with higher heterogeneity, and acknowledging this in prediction intervals suggested that results are inconclusive for memantine.

We comment on this in the discussion section (see underlined):

Lines 352-360: "For patients with mild to moderate cognitive impairment, all except for memantine and its combination with transdermal rivastigmine, were associated with increased odds of a SAE, yet none of these results reached statistical significance. Overall, memantine was associated with lower odds of a SAE than placebo, yet this was statistically significant only in the subnetwork analysis including aggregate data (i.e., studies without IPD) and the meta-regression analysis using study duration as a covariate. However, acknowledging for heterogeneity in the network, prediction intervals suggested that results are inconclusive and the odds of SAE could not be differentiated between memantine and placebo. Of note, the accuracy of SAE reporting may be impacted by the degree of cognitive impairment."

14. It is not clear from the objectives and methods of the review why disease severity has been selected from among other potential effect modifiers for particular emphasis in the discussion and in the abstract and this should be explained. It is of clinical interest, but was it a post hoc decision?

Authors' response: As already mentioned in our protocol (see relevant text below) we a priori planned to explore AD severity, and requested this information from all IPD providers. We highlighted this in the introduction section and presented relevant details for our IPD analysis in Appendix 1. Disease severity is of particular clinical interest and our results showed that there can be differences in the efficacy of different cognitive enhancers, which we discuss in the main text. Assessment of other potential effect modifiers (e.g., comorbidities) did not reveal a clear picture on how the efficacy or safety of cognitive may change, and relevant results are presented in the supplementary file.

Protocol: "In addition, in our previous NMA, we attempted a subgroup analysis for AD severity, but we were unable to infer on the treatment effectiveness for the severe AD subgroup because there were only few RCTs available that reported on patients with severe AD and a NMA was impossible (disconnected network). The advantage of IPD is that we are not restricted to using the information reported in the publication. For example, for the 15 RCTs that did not report severity of disease in patients, we will be able to include them in the IPD-NMA analysis. Also, we will be able to use the information on severe AD from studies that included patients ranging from mild-to-severe and moderate-to-severe disease."

Introduction, Lines 74-76: "In AD, disease severity and sex are potential effect modifiers. However, aggregate data and covariates of interest (e.g., sex, disease severity) are not consistently reported across randomized clinical trials (RCTs).⁴"

15. There is a good discussion of the limitations of the included studies, but insufficient discussion of other, clinically important limitations of the review, including utility and consistency of definition of outcome measures, the absence of a well-established MCID for the MMSE, and lack of consideration of drug doses. As a result, I find it difficult to draw clinically applicable implications from this review of the data. The authors could consider ways to enhance clinical applicability, e.g. by ranking efficacy of defined doses of drugs for clearly defined disease severity categories and by being very explicit about how certain they are of any meaningful differences between the treatments.

Authors' response: Thank you for this comment. We added these limitations in the discussion section as shown below:

Lines 434-439: "Sixth, there are clinically important limitations associated with this review, including consistent definition of outcome measures across studies, a well-established MCID for the MMSE score, lack of consideration of drug doses due to inconsistent reporting and data availability bias that we were unable to overcome (15% of the studies shared their IPD). Future studies are needed to establish ranking efficacy in drug doses and combination of interventions across different disease severity categories."

VERSION 2 – REVIEW

REVIEWER	Peter Watson University of Cambridge, MRC Cognition and Brain Sciences Unit
REVIEW RETURNED	16-Sep-2021
GENERAL COMMENTS	Comparative safety and efficacy of cognitive enhancers for Alzheimer's dementia: A systematic review with individual patient data network meta-analysis bmjopen-2021-053012.R1 My main concern is that the prediction intervals used in these meta-analyses, as admitted by the authors, suggest there is large variation among individuals leading to inconclusive results pertaining

	to the effectiveness, or otherwise, of the inhibitors in this study. Page 4, lines 20-23 and page 11, line 225. The abstract suggests Donepezil on its own and with memantine improves MMSE scores yet on page 11, line 225 where these results are presented in the text that in addition the Prediction intervals (PIs) "are not conclusive" which appears to undermine these results and is also omitted from the abstract. Page 5, lines 58-59. A limitation of the meta-analyses is a high risk of bias due to attrition. This seems serious to me in giving an incomplete view of the usefulness of the inhibitors in general populations since only a small subset of people stay in the studies examined. The main reason for these high rates of dropouts is "experiencing an adverse event" (page 10, lines 194-195). This sounds rather serious to me if it is occurring in the high numbers which are implied and I wondered if this undermines the usefulness of these inhibitors and what sort of people suffer adverse effects and what these adverse effects actually are. On ethical grounds you might want to screen for characteristics known to increase the chance of such adverse reactions prior to administering these inhibitors. I also note that two-thirds of studies (page 10, lines 188-189) were potentially at risk of funding bias. No further details are given but given its large influence on the type of studies in this paper it might be worth explaining this type of bias further. You don't mention and, to try and deal with bias, if dropout was accounted for in the studies examined in this paper e.g. through using random effects models? Page 8, lines 124-126. I assume a fixed effect model was used as opposed to a random effect model due to lack of between study heterogeneity? Page 8, line 145. There are multiple references in the paper to P-scores. Are these simply p-values usually denoted as 'p='? You mention using P-scores on page 8, line 145 without further comment so I assumed these are the same as p-values? I am not sure these are P-values however since, for example, on page 14, lines 314-316 you mention statistical significance for memantine and moreover quote an odds ratio whose 95% confidence interval does not contain the null odds ratio of '1' indicating no difference between memantine and placebo yet you have a P-score of 88% which, if it was a p-value, would suggest, contrarily, there was no difference between the treatments. I am not, therefore, clear what you base your assertion of statistical significance on - is it the confidence interval not containing a null effect (e.g. zero difference) or an (unquoted) p-value? Page 11, lines 223 to page 12, line 248. You don't include P-values/P-scores in the comparisons here for the NMA comparing IPD and aggregate data together and separately yet these are included in the additional analyses on page 12, lines 252 to 271. To be consistent you should include P-scores for all comparisons or explain why they are omitted. Page 11, lines 225-229. You mention here that a range of pairwise comparisons of mean differences (presumably all with placebo?) are associated with a 'minimal clinically important difference' which you quote as 1.40 yet the 95% confidence intervals you quote for these
--	--

mean differences include differences which are quite a bit less than this and in some cases (negative mean differences) appear to benefit the placebo. I also suspect that since the 95% Confidence Intervals include the zero mean difference that P-values for these comparisons would not be statistically significant which would lead to a conclusion of no difference between any of these treatments and placebo. There is also a mention on line 225 on page 11 for studies involving Donepezil on its own and with another inhibitor that "PIs suggested results are not conclusive" which appears to undermine your results on lines 223-225 suggesting they are superior to placebo on MMSE score with confidence intervals that do not contain a zero mean difference. This inconclusivity suggesting doubt about the usefulness of inhibitors in improving MMSE is also mentioned on page 12, line 261 using prediction intervals which imply a wide range of changes using the inhibitors. No figures are quoted here for the ranges of these Prediction Intervals but it may be that the prediction intervals suggest that using inhibitors could lead to possibly detrimental effects. I would explain on page 8 that a prediction interval, unlike a confidence interval, is the predicted difference for a new person coming into the study so people can better understand the results. I also would then explain how a prediction interval can show one thing (no evidence of differences) whereas a confidence interval can show a difference so we can better understand the apparent contradictions in the results. I think you could mention that the prediction interval adds an extra component of variance to explain for individual variation whereas a confidence interval looks at overall values and, as such the former is always larger. One could argue that we should only focus on prediction intervals since these inhibitors are predominantly going to be used on people who are new to the study and who, as a result, would have this extra uncertainty.

Page 12, lines 263-271. The use of both confidence intervals and predictor intervals leads to a confusing description of the results here. For example on lines 265-268 on page 12 it states that oral rivastigmine improves MMSE but that results are inconclusive due to 95% Prediction Intervals including a zero difference indicating no change using rivastigmine.

Page 15, lines 334-345, 343-344 and 354. I think the results are saying that it may be that for some people these inhibitors may be beneficial but it is not clear to me which people these would be ie what characteristics of these people would yield a beneficial outcome using the inhibitors in this paper. You mention this on lines 343-344 on page 15 in the discussion. I, therefore, agree with the statement there and would emphasise that one has to be careful about suggesting the use of inhibitors when one does not know what sort of effect they will have on the individual. In the discussion (page 15, lines 334-335) you also correctly mention that there is heterogeneity across the individuals which leads to substantially larger predictor intervals which reinforces this point about knowing which characteristics of the individual influence the effectiveness of the inhibitors on, say, MMSE. I am not convinced that this paper, worthy though these analyses are, however actually adds anything positive to what seems to be already known. The authors attempt to motivate use of inhibitors based upon patients' degree of cognitive impairment but they conclude "none of the results reach statistical significance" on line 354 on page 15. In a nutshell results on whether inhibitors are beneficial are inconclusive. I think you need to further motivate what characteristics of an individual influence the

	effect of inhibitors so that clinicians can with confidence use these on identified subsets of people. Page 29, Figure 3. I think you mean to say here these forest plots include all listed study types denoted i) to iv) e.g. aggregate data, AD and crude results from IPD studies as there are just two plots for MMSE and SAE rather than a single plot for each study type. Page 30, Figure 4. I don't find the heat plot here informative and don't think it adds anything to the results of the meta-analyses described in the text. You have presented the relevant meta-analyses where you can add in P-scores in the text and (also in Appendix 16) and these should be sufficient.
--	---

REVIEWER	Jenny McCleery Oxford Health NHS Foundation Trust, Mental Health
REVIEW RETURNED	22-Sep-2021

GENERAL COMMENTS	I thank the authors for their responses to my earlier comments. However, two issues related to the interpretation of the results still cause me concern. 1. The issue of minimum clinically important difference (MCID) on the MMSE. The authors, in their response to other comments, explain that they do not dichotomise results into significant and non-significant on the basis of a P value. However, it is equally unhelpful to dichotomise results in the abstract conclusions (or elsewhere) as either clinically important for cognition or not on the basis of a mean difference from placebo of greater than or less than 1.4 MMSE points. (Note: abstract refers to "MMSE score greater than 1.4" when it means "MD from placebo of 1.4 MMSE points"). I would emphasise again that 1.4 points is not a well-established MCID but was estimated in one study among participants with severe dementia only. It is not appropriate to imply that it is an accepted figure or to use it to interpret results as clinically important or otherwise across the whole severity range. 2. The issue of definition of the SAE outcome. I understand that adverse events, even more than efficacy outcomes, are inconsistently reported and this is clearly a problem for systematic reviewers. The authors point to the definitions of SAEs in Appendix 19. However, the definitions given in the Appendix are not all definitions of what was considered an SAE in each study; some are simply descriptions of the AE coding methods used, and some refer to any adverse event. If a study did not report specifically on SAEs, but did report data on 'any adverse event', were these data included or excluded? It appears that they may have been included in some cases, which should be commented on in relation to the study inclusion criteria. If drawing conclusions about SAEs, then the authors should consider conducting sensitivity analyses confined to those studies which report data on what are commonly understood to be SAEs, i.e. death, life-threatening events, hospitalisation, serious disability. Otherwise, conclusions which purport to be about SAEs risk being misleading. A typographical error which recurs a couple of time is "drug regiments" instead of "regimens".
---

VERSION 2 – AUTHOR RESPONSE

Reviewer 1 Comments:

1. My main concern is that the prediction intervals used in these meta-analyses, as admitted by the authors, suggest there is large variation among individuals leading to inconclusive results pertaining to the effectiveness, or otherwise, of the inhibitors in this study.

Authors' response: Thank you for this comment. We agree that the calculated prediction intervals (PIs) in our review are very wide, which particularly highlights the considerable amount of heterogeneity in our findings. As also highlighted in the Cochrane Handbook, the confidence interval (CI) from a random-effects model shows the uncertainty in the summary effect, i.e. the mean of the study-specific effects, but it does not show the degree of heterogeneity. PIs are a popular way of expressing the amount of heterogeneity, and Salanti and colleagues (<https://pubmed.ncbi.nlm.nih.gov/32243458/>) propose the use of PIs to facilitate the assessment of heterogeneity for each intervention comparison (this is also part of the credibility assessment in the NMA results using CINeMA). Hence, we prefer to keep PIs in our systematic review. In the absence of heterogeneity, confidence intervals and prediction intervals are equal. However, PIs are based on the assumption of a normal distribution for the study-specific effects, and they may be problematic in the presence of funnel plot asymmetry. To this end, we have commented on this in the Supplementary file 1, Appendix 1, as shown below:

“CINeMA assesses the credibility of the NMA results and heterogeneity examining the range of both confidence intervals (CIs; which do not capture heterogeneity) and prediction intervals (PIs; which capture heterogeneity) in relation to their equivalence. If a PI includes values that lead to a different conclusion than an assessment based on the corresponding CI, then this suggests that there is considerable heterogeneity. PIs are expected to include the true intervention effects in future studies with characteristics similar to the existing studies, and they incorporate the extent of between-study heterogeneity.^{5 6} In the presence of considerable heterogeneity, they are wide to include intervention effects with different implications for practice. However, caution is needed in the interpretation of results in the presence of funnel plot asymmetry, since PIs are based on the assumption of a normal distribution for the study-specific effects and as such, they may be problematic if the data do not follow a normal distribution.”

2. Page 4, lines 20-23 and page 11, line 225. The abstract suggests Donepezil on its own and with memantine improves MMSE scores yet on page 11, line 225 where these results are presented in the text that in addition the Prediction intervals (PIs) "are not conclusive" which appears to undermine these results and is also omitted from the abstract.

Authors' response: Thank you for this remark. We decided to remove the presentation of PIs from the main text in the MMSE outcome due to evidence of funnel plot asymmetry in the comparison-adjusted funnel plot. Although PIs show the presence of important heterogeneity, their interpretation in such a case can be problematic. We present the PI results for both outcomes in the Supplementary file 1, to exemplify the interpretation of the CINeMA assessment and imprecision due to heterogeneity. We also noted this in the discussion section:

Lines 341-347, “However, heterogeneity was a major concern, which requires careful consideration before suggesting the use of cognitive enhancers, and particularly when the efficacy is not clear on the patient’s characteristics. This was also captured by PIs, but their interpretation requires caution due to evidence of funnel plot asymmetry in the MMSE outcome. Overall, PIs are expected to include the true intervention effect expected in future studies, and they incorporate an extra component of variance, specifically between-study heterogeneity. In the absence of heterogeneity, confidence intervals and PIs are equal.”

3. Page 5, lines 58-59. A limitation of the meta-analyses is a high risk of bias due to attrition. This seems serious to me in giving an incomplete view of the usefulness of the inhibitors in general populations since only a small subset of people stay in the studies examined. The main reason for these high rates of dropouts is "experiencing an adverse event" (page 10, lines 194-195). This sounds rather serious to me if it is occurring in the high numbers which are implied and I wondered if this undermines the usefulness of these inhibitors and what sort of people suffer adverse effects and what these adverse effects actually are. On ethical grounds you might want to screen for characteristics known to increase the chance of such adverse reactions prior to administering these inhibitors. I also note that two-thirds of studies (page 10, lines 188-189) were potentially at risk of funding bias. No further details are given but given its large influence on the type of studies in this paper it might be worth explaining this type of bias further.

Authors' response: We agree that attrition bias is a concern of this review and we have highlighted this in the limitations of our discussion section. Unfortunately, we do not have access to the IPD anymore, which would allow further exploration. Access to the IPD of the included studies was only for a certain amount of time. We agree with the reviewer that the high rates of dropouts due to experiencing an adverse event is a concern and the patients' characteristics that may increase the chances of such adverse reactions prior to administering these cognitive enhancers should further be explored. We updated our discussion accordingly:

Lines 430-436: "Second, missing data is a big concern in the published RCTs for AD. We found high rates of dropouts from experiencing an adverse event and the patients' characteristics that may increase the chances of such adverse reactions prior to administering these cognitive enhancers should further be explored. To assess the impact of missing data in our NMA, we applied the informative missingness of difference in means.³⁰ However, future studies should explore the characteristics of missing participants and specific adverse events."

Regarding funding bias, included in the "Other" item of the risk of bias assessment, we added an explanation on our judgements in Appendix 8, as an asterisk to the table:

****** Other bias was categorized as:***

a) low risk of bias when the study appeared to be free of other sources of bias,

b) high risk of bias when there was at least one important risk of bias. For example, when the study had:

- A potential source of bias related to the specific study design used; or***
- A conflict of interest related to funding source; or***
- An author was an employee of the drug company that sponsored the study; or***
- Been claimed to have been fraudulent; or***
- Other potential biases.***

c) unclear risk of bias when there was a potential for bias, but there was either:

- Insufficient information to assess whether an important risk of bias exists; or***
- Insufficient rationale/evidence that an identified problem would introduce bias;***

or

- Funding by drug company, but conflicts were not described"***

Also, in Table 1 we present the study categorization according to their funding details, and the 60% of the included studies were industry funded, whereas all the IPD retrieved from the data sharing platforms and sponsors were industry sponsored. We commented on the potential for funding bias in the discussion section:

Lines 443-447: “Fourth, the comparison-adjusted funnel plot for MMSE suggested there is an indication for small-study effects pointing to the treatment being better, and results should be interpreted with caution. This may also be related to the potential risk of funding bias, since the majority of the included studies were industry-sponsored and IPD were retrieved only from industry-sponsored studies favouring cognitive enhancers over placebo.”

4. You don't mention and, to try and deal with bias, if dropout was accounted for in the studies examined in this paper e.g. through using random effects models?

Authors' response: *To evaluate the impact of missing data in our NMA, we applied the 'informative missingness difference of means' (IMDoM) imputation method for MMSE on the aggregate data level. See Supplementary file 1, Appendices 1 and 16. We could not perform a multiple imputation technique on the IPD level since the data sharing platforms did not allow us to download the required R packages to perform the relevant analyses. We also report this as one of the encountered challenges to conduct this systematic review with IPD-NMA in Appendix 21:*

“Software availability: Required R packages (e.g., mice) were not available/provided, and we were not allowed to install any new R packages; some R packages were older versions (e.g. lme4).”

5. Page 8, lines 124-126. I assume a fixed effect model was used as opposed to a random effect model due to lack of between study heterogeneity?

Authors' response: *To assess for evidence of small-study effects using the 'comparison-adjusted' funnel plot, we used the fixed effect model for the standard meta-analysis effect for each treatment comparison, ordered treatments chronologically according to year of availability in Canada, and presented only treatment comparisons versus placebo. We decided to use the fixed effect model for the standard meta-analysis conducted for each treatment comparison in this plot to better assess for evidence of small-study effects. This is because random effects give higher weight to small studies and in the presence of small-study effects, the summary effect would be biased towards one direction. Funnel plot asymmetry would denote that there are differences between the estimates derived from small and large studies, and the efficacy between active treatments and placebo.*

We added more details regarding the conduct of the comparison-adjusted funnel plot in Appendix 1:

“We assessed reporting bias based on the comparison-adjusted funnel plot since there are no established statistical methods to explore reporting bias. We used a comparison-adjusted funnel to account for the fact that each set of studies estimates a different summary effect in NMA. This is a scatterplot of the difference between the study-specific effect sizes from the corresponding comparison-specific effect (obtained from standard meta-analysis) against the corresponding study-specific standard error. We used the fixed effect model for the standard meta-analysis performed for each treatment comparison, ordered treatments chronologically according to year of availability in Canada, and used only treatment comparisons versus placebo. We used the netfunnel command in Stata to produce the comparison-adjusted funnel plot.⁴”

6. Page 8, line 145. There are multiple references in the paper to P-scores. Are these simply p-values usually denoted as 'p='? You mention using P-scores on page 8, line 145 without further comment so I assumed these are the same as p-values? I am not sure these are P-values however since, for example, on page 14, lines 314-316 you mention statistical significance for memantine and moreover quote an odds ratio whose 95% confidence interval

does not contain the null odds ratio of '1' indicating no difference between memantine and placebo yet you have a P-score of 88% which, if it was a p-value, would suggest, contrarily, there was no difference between the treatments. I am not, therefore, clear what you base your assertion of statistical significance on - is it the confidence interval not containing a null effect (e.g. zero difference) or an (unquoted) p-value?

Authors' response: Treatment ranking measures, such as SUCRAs and P-scores, are popular tools for quantifying the treatment performance in NMA, and is a standard process to rank the safety and efficacy of the included interventions. We refer the reviewer to our cited publications [17, 18] as well as the Cochrane Handbook and Chapter 11 for more details on these statistics. We also added the following paragraph in the supplementary file 1 and appendix 1:

"We present the results using summary effect sizes, and in particular the MD for MMSE and the OR for AE, along with their corresponding CIs and PIs.⁶ We ranked the interventions for each outcome according to their efficacy and safety using P-scores in frequentist analyses and SUCRAs (surface under the cumulative ranking curve) in Bayesian analyses (e.g., meta-regression analysis).²²⁻²³ SUCRA is the numeric presentation of the intervention ranking and is based on the surface under the cumulative ranking probability function for each treatment. An equivalent frequentist statistic is the P-score measure that is based on the observed treatment effect estimates and their uncertainty. Both measures summarize the estimated probabilities for all possible ranks, account for uncertainty in relative ranking, and range between 0-100%, with 100% reflecting the best intervention with no uncertainty and 0% reflects the worst intervention with no uncertainty. Ranking strategies are commonly encountered in NMAs,²⁴⁻²⁶ and we present the hierarchy of cognitive enhancers in a rank-heat plot.^{27"}

7. Page 11, lines 223 to page 12, line 248. You don't include P-values/P-scores in the comparisons here for the NMA comparing IPD and aggregate data together and separately yet these are included in the additional analyses on page 12, lines 252 to 271. To be consistent you should include P-scores for all comparisons or explain why they are omitted.

Authors' response: This is an incredibly large and complex systematic review and NMA. Despite the plethora of data and results, our aim is to be transparent and present everything we obtained from our multiple analyses. We attempt to present all information in our manuscript, but considering also that we are over the word limit (4215 total words in contrast to 4,000 words), we present key results in the text and the remaining results in the included tables, figures, and appendices. In particular, we present the P-scores for all treatments included in our NMA in a rank-heat plot in Figure 4.

8. Page 11, lines 225-229. You mention here that a range of pairwise comparisons of mean differences (presumably all with placebo?) are associated with a 'minimal clinically important difference' which you quote as 1.40 yet the 95% confidence intervals you quote for these mean differences include differences which are quite a bit less than this and in some cases (negative mean differences) appear to benefit the placebo. I also suspect that since the 95% Confidence Intervals include the zero mean difference that P-values for these comparisons would not be statistically significant which would lead to a conclusion of no difference between any of these treatments and placebo. There is also a mention on line 225 on page 11 for studies involving Donepezil on its own and with another inhibitor that "PIs suggested results are not conclusive" which appears to undermine your results on lines 223-225 suggesting they are superior to placebo on MMSE score with confidence intervals that do not contain a zero mean difference. This inconclusivity suggesting doubt about the usefulness of inhibitors in improving MMSE is also mentioned on page 12, line 261 using prediction intervals which imply a wide range of changes using the inhibitors. No figures are quoted here for the ranges of these Prediction Intervals but it may be that the prediction intervals suggest that using inhibitors could lead to possibly detrimental effects. I would explain on page 8 that a prediction interval, unlike a confidence interval, is the predicted difference for a new person coming into the study so people can better understand the results. I also would then explain how a prediction interval can show one thing (no evidence of differences) whereas a confidence interval can show a difference so we can better understand the apparent

contradictions in the results. I think you could mention that the prediction interval adds an extra component of variance to explain for individual variation whereas a confidence interval looks at overall values and, as such the former is always larger. One could argue that we should only focus on prediction intervals since these inhibitors are predominantly going to be used on people who are new to the study and who, as a result, would have this extra uncertainty.

Authors' response: Thank you for raising these points. We have updated the text interpreting the results relevant to minimally clinical important difference (MCID) as shown below:

Lines 226-233: "Transdermal rivastigmine (MD= 2.11, 95% CI: -0.04 to 4.26), and the combinations donepezil+memantine, galantamine+memantine (MD= 2.24, 95% CI: -2.13 to 6.61), and transdermal rivastigmine+memantine (MD= 1.79, 95% CI: -1.70 to 5.27) were associated with a MD from placebo of more than 1.40 MMSE points. A previous study suggested a MD larger than 1.40 is a minimal clinically important difference (MCID).²¹ However, the associated 95% CIs were quite imprecise spanning between a mean decrease below and a mean increase above the suggested MCID value (Figure 3a)."

Regarding the interpretation of PIs, as per our response to comment #1, we added more information in the Supplementary file 1, Appendix 1. Also, provided that there was evidence of funnel plot asymmetry in the comparison-adjusted funnel plot of the MMSE outcome, we decided to not base the interpretation of our findings on PIs. However, PIs were used in the assessment of imprecision due to heterogeneity in the CINeMA tool. Please see also our response to comment #2.

9. Page 12, lines 263-271. The use of both confidence intervals and predictor intervals leads to a confusing description of the results here. For example on lines 265-268 on page 12 it states that oral rivastigmine improves MMSE but that results are inconclusive due to 95% Prediction Intervals including a zero difference indicating no change using rivastigmine.

Authors' response: Please see response in comment #2.

10. Page 15, lines 334-345, 343-344 and 354. I think the results are saying that it may be that for some people these inhibitors may be beneficial but it is not clear to me which people these would be ie what characteristics of these people would yield a beneficial outcome using the inhibitors in this paper. You mention this on lines 343-344 on page 15 in the discussion. I, therefore, agree with the statement there and would emphasise that one has to be careful about suggesting the use of inhibitors when one does not know what sort of effect they will have on the individual. In the discussion (page 15, lines 334-335) you also correctly mention that there is heterogeneity across the individuals which leads to substantially larger predictor intervals which reinforces this point about knowing which characteristics of the individual influence the effectiveness of the inhibitors on, say, MMSE. I am not convinced that this paper, worthy though these analyses are, however actually adds anything positive to what seems to be already known. The authors attempt to motivate use of inhibitors based upon patients' degree of cognitive impairment but they conclude "none of the results reach statistical significance" on line 354 on page 15. In a nutshell results on whether inhibitors are beneficial are inconclusive. I think you need to further motivate what characteristics of an individual influence the effect of inhibitors so that clinicians can with confidence use these on identified subsets of people.

Authors' response: As per our response to comment #2, we updated our discussion including the following sentence:

Lines 341-343: "However, heterogeneity was a major concern, which requires careful consideration before suggesting the use of cognitive enhancers, and particularly when the efficacy is not clear on the patients' characteristics."

11. Page 29. Figure 3. I think you mean to say here these forest plots include all listed study types denoted i) to iv) e.g. aggregate data, AD and crude results from IPD studies as there are just two plots for MMSE and SAE rather than a single plot for each study type.

Authors' response: *In Figure 3 we present the summary results obtained from each NMA analysis for each treatment comparison against placebo. We include the results of the NMA that included*

- i. aggregate data (AD) and fully adjusted results from studies with available individual patient data (IPD),*
- ii. AD and crude results from studies with available IPD,*
- iii. AD only (studies with available IPD are not included in the analysis), and*
- iv. crude results from individual studies with individual patient data (IPD)*

For more details on our analyses, we refer the reviewer to the methods section and the supplementary file 1, appendix 1.

12. Page 30, Figure 4. I don't find the heat plot here informative and don't think it adds anything to the results of the meta-analyses described in the text. You have presented the relevant meta-analyses where you can add in P-scores in the text and (also in Appendix 16) and these should be sufficient.

Authors' response: *We would prefer to keep the rank-heat plot for the MMSE outcome presenting the ranking statistics across our different NMA analyses in the main manuscript to improve reporting and transparency of our findings. Including the rank-heat plot in our manuscript helps with reducing the word count in the text. Also, the rank-heat plot has been well accepted by multiple systematic reviewers and has 90 citations the last 5 years (source: Google Scholar). We have also received feedback from clinicians (e.g. St. Michael's Hospital Unity Health Toronto Acute Care of the Elderly team) that rank-heat plots facilitate results interpretation. See:*

<https://pubmed.ncbi.nlm.nih.gov/32546202/>

<https://pubmed.ncbi.nlm.nih.gov/31610547/>

Reviewer 2 Comments:

I thank the authors for their responses to my earlier comments. However, two issues related to the interpretation of the results still cause me concern.

1. The issue of minimum clinically important difference (MCID) on the MMSE. The authors, in their response to other comments, explain that they do not dichotomise results into significant and non-significant on the basis of a P value. However, it is equally unhelpful to dichotomise results in the abstract conclusions (or elsewhere) as either clinically important for cognition or not on the basis of a mean difference from placebo of greater than or less than 1.4 MMSE points. (Note: abstract refers to "MMSE score greater than 1.4" when it means "MD from placebo of 1.4 MMSE points"). I would emphasise again that 1.4 points is not a well-established MCID but was estimated in one study among participants with severe dementia only. It is not appropriate to imply that it is an accepted figure or to use it to interpret results as clinically important or otherwise across the whole severity range.

Authors' response: *Thank you for this comment. We agree that interpretation should not be based on categorization of results using statistical or clinical significance, and have updated the relevant text in the manuscript. Please note that we decided to keep the MCID of 1.40 MMSE points, based on a previous publication [see #21 in main manuscript], to ease*

interpretation of MD results and to assess credibility in our NMA results as required in the CINEMA approach.

Abstract, lines 34-37: “The MDs of all cognitive enhancer regimens except for single-agent oral rivastigmine, galantamine, and memantine, against placebo were clinically important for cognition (MD larger than 1.40 MMSE points), but results were quite imprecise.”

Lines 226-233: “Transdermal rivastigmine (MD= 2.11, 95% CI: -0.04 to 4.26), and the combinations donepezil+memantine, galantamine+memantine (MD= 2.24, 95% CI: -2.13 to 6.61), and transdermal rivastigmine+memantine (MD= 1.79, 95% CI: -1.70 to 5.27) were associated with a MD from placebo of more than 1.40 MMSE points. A previous study suggested a MD larger than 1.40 is a minimal clinically important difference (MCID).; above 1.40)²¹ However, the associated 95% CIs were quite imprecise spanning between a mean decrease below and a mean increase above the suggested MCID (Figure 3a).”

Lines 242-244: “Assuming an MCID of 1.40, results were in agreement with the NMA of IPD and aggregate data, and donepezil+memantine (MD= 2.71, 95% CI: -0.17 to 5.60) was likely the most effective in improving MMSE score (P-score= 76%).”

Lines 249-251: “Donepezil (MD= 0.70, 95% CI: 0.01 to 1.40) and transdermal rivastigmine (MD= 1.06, 95% CI: 0.04 to 2.08) were superior to placebo, but none of the point estimates reached a previously suggested MCID.²¹”

Lines 377-380: “Considering a MCID equal to 1.40 points,²¹ the MDs of all cognitive enhancer regimens except for single-agent oral rivastigmine, galantamine, and memantine, against placebo were clinically important for cognition, but these were associated with high uncertainty. However, the 1.40 MMSE cut-off value is not a widely adopted MCID.”

Lines 400-403: “The only difference was observed in transdermal rivastigmine that was associated with a MCID of greater than 1.40 MMSE points against placebo in the aggregate data NMA compared to the IPD NMA, yet a statistically significant improvement was achieved in the IPD NMA.”

Lines 455-459: “Sixth, there are clinically important limitations associated with this review, including consistent definition of outcome measures across studies, a well-established MCID for the MMSE score, lack of consideration of drug doses due to inconsistent reporting and data availability bias that we were unable to overcome (15% of the studies shared their IPD).”

Lines 296-301: “According to P-score, oral rivastigmine had the least favourable safety profile regarding AE (OR= 1.26, 95% CI: 0.82 to 1.94, P-score= 16%), followed by donepezil (OR= 1.08, 95% CI: 0.87 to 1.35, P-score= 30%) and galantamine+memantine (OR= 1.03, 95% CI: 0.45 to 2.39, P-score= 43%), yet in none of these comparisons the odds of experiencing a AE were imprecise and not importantly statistically significant different from placebo (Figure 3b; Appendices 16, 18).”

Lines 306-310: “Results were mainly consistent with NMA of IPD and aggregate data, but for memantine which was 0.70 times less likely to experience a AE than placebo, with an OR ranging from statistically significantly associated with lower odds of a AE than placebo when using aggregate data only (OR 0.70, 95% CI: 0.51 to 0.97, (P-score= 77%).)”

Lines 320-323: “Additional analyses using both IPD and aggregate data, showed that memantine was 0.61 times less likely to experience a AE than placebo when using study duration as a covariate, with an OR ranging from 0.37 to 0.93(P-score= 88%).”

2. The issue of definition of the SAE outcome. I understand that adverse events, even more than efficacy outcomes, are inconsistently reported and this is clearly a problem for systematic reviewers. The authors point to the definitions of SAEs in Appendix 19. However, the definitions given in the Appendix are not all definitions of what was considered an SAE in

each study; some are simply descriptions of the AE coding methods used, and some refer to any adverse event. If a study did not report specifically on SAEs, but did report data on 'any adverse event', were these data included or excluded? It appears that they may have been included in some cases, which should be commented on in relation to the study inclusion criteria. If drawing conclusions about SAEs, then the authors should consider conducting sensitivity analyses confined to those studies which report data on what are commonly understood to be SAEs, i.e. death, life-threatening events, hospitalisation, serious disability. Otherwise, conclusions which purport to be about SAEs risk being misleading.

Authors' response: Thank you for pointing this out. We reviewed the study definitions and updated Appendix 19 wherever needed. We decided to use the term adverse events (AE) instead of SAE for a more relevant representation of results, and since most articles did not categorize by severity of adverse events. We updated the manuscript, Appendices, and plots accordingly. We also added the following sentence to the Supplementary file 1, Appendix 1:

"We considered an adverse event (AE) as defined in the individual trials. Definitions were captured for each study separately."

3. A typographical error which recurs a couple of times is "drug regiments" instead of "regimens".

Authors' response: Done. This has now been corrected.

VERSION 3 – REVIEW

REVIEWER	Peter Watson University of Cambridge, MRC Cognition and Brain Sciences Unit
REVIEW RETURNED	01-Feb-2022

GENERAL COMMENTS	Comparative safety and efficacy of cognitive enhancers for Alzheimer's dementia: A systematic review with individual patient data network meta-analysis bmjopen-2021-053012.R2 I think the authors have fleshed out the script adding in important information and I am happy with the script. The limitations of the studies they look at are flagged in the discussion including bias of using LOCF for missing values. I think the important limitation which the authors flag on page 16, line 374 is the high variability in the size of the benefits of the treatments. On a minor level I wondered if this high variability in effect sizes may be related to different populations of people in the studies with perhaps some nationalities or genetical profiles benefitting from the treatments more consistently than others and more likely to show clinically meaningful improvements? Such knowledge would enable a more efficient targeted approach to using these treatments in the future. I like the fact the authors have checked their results across age, sex and other medications (page 16, line 376). This at least suggests that there is no evidence to suggest any imbalances in these variables between placebo and treatment groups so that any benefits of the treatments were not due to some groups having, say, a higher proportion of older people in them although the authors honestly admit the low power of these tests which may mean differences were there but not detected.
---

REVIEWER	Jenny McCleery Oxford Health NHS Foundation Trust, Mental Health
-----------------	---

REVIEW RETURNED	01-Dec-2021
GENERAL COMMENTS	I have no further comments for the authors.

VERSION 3 – AUTHOR RESPONSE

Reviewer 1 Comments:

1. I think the authors have fleshed out the script adding in important information and I am happy with the script.

Authors' response: We thank the reviewer for their time spending on reviewing our manuscript and their helpful feedback.

2. The limitations of the studies they look at are flagged in the discussion including bias of using LOCF for missing values. I think the important limitation which the authors flag on page 16, line 374 is the high variability in the size of the benefits of the treatments. On a minor level I wondered if this high variability in effect sizes may be related to different populations of people in the studies with perhaps some nationalities or genetical profiles benefitting from the treatments more consistently than others and more likely to show clinically meaningful improvements? Such knowledge would enable a more efficient targeted approach to using these treatments in the future.

Authors' response: The reviewer raises a very interesting point. Knowledge of the different populations included in the studies, such as genetic profiles, race, and gender identity, would enable exploration of which population characteristics would benefit more, with regard to clinically important improvements, when using the treatments. However, this information was rarely reported in our included studies. We added a sentence about this in the Discussion section:

Lines 378-385: "Considering a MCID equal to 1.40 points,²¹ the MDs of all cognitive enhancer regimens except for single-agent oral rivastigmine, galantamine, and memantine, against placebo were clinically important for cognition, but these were associated with high uncertainty. However, the 1.40 MMSE cut-off value is not a widely adopted MCID. Also, high variability may be related to different populations included in the studies, such as genetic profiles, race, and gender identity. Future studies should report this information to enable exploration of population characteristics that would benefit more, with a clinically important improvement, when using these treatments."

3. I like the fact the authors have checked their results across age, sex and other medications (page 16, line 376). This at least suggests that there is no evidence to suggest any imbalances in these variables between placebo and treatment groups so that any benefits of the treatments were not due to some groups having, say, a higher proportion of older people in them although the authors honestly admit the low power of these tests which may mean differences were there but not detected.

Authors' response: We agree with the reviewer and we thank them for their kind words.

Reviewer 2 Comments:

I have no further comments for the authors.

Authors' response: We thank the reviewer for their time spending on reviewing our manuscript and their helpful feedback.